



# A detailed characterization of the Saharan dust collected during the Fennec Campaign in 2011: *in situ* ground-based and laboratory measurements

Adriana Rocha-Lima[1,2], J. Vanderlei Martins[1,3], Lorraine A. Remer[1], Martin Todd[4], John H. Marsham[5,6], Sebastian Engelstaedter[7], Claire L. Ryder[8], Carolina Cavazos-Guerra[9], Paulo Artaxo[10], Peter Colarco[2], and Richard Washington[7]

[1]University of Maryland, Baltimore County, Baltimore, MD, United State
[2]Atmospheric Chemistry and Dynamic Laboratory, NASA Goddard Space Flight Center, Greenbelt, MD, United States
[3]Climate and Radiation Laboratory, NASA Goddard Space Flight Center, Greenbelt, MD, United States
[4]Department of Geography, University of Sussex, UK
[5]School of Earth and Environment, University of Leeds, Leeds, UK
[6]National Center for Atmospheric Science, UK
[7]Climate Research Lab, Oxford University Center for the Environment, Oxford, UK
[8]Department of Meteorology, University of Reading, UK
[9]Institute for Advanced Sustainability Studies, Potsdam, Germany
[10]Instituto de Física, Universidade de São Paulo, São Paulo, Brazil

*Correspondence to:* Adriana Rocha-Lima (limadri1@umbc.edu)

**Abstract.** Millions of tons of mineral dust are lifted by the wind from arid surfaces and transported around the globe every year. The physical and chemical properties of the mineral dust are needed to better constrain remote sensing observations and are of fundamental importance for the understanding of dust atmospheric processes. Ground-based *in situ* measurements and *in situ* filter collection of Saharan dust were obtained during the Fennec campaign in the central Sahara in 2011. This paper

presents results of the absorption and scattering coefficients, and hence, single scattering albedo (SSA), of the Saharan dust measured in real time during the last period of the campaign, and subsequent laboratory analysis of the dust samples collected in two supersites, SS1 and SS2, in Algeria and in Mauritania, respectively. The samples were taken to the laboratory where their size and aspect ratio distributions, mean chemical composition, spectral mass absorption efficiency and spectral imaginary refractive index were obtained from the ultraviolet (UV) to the near infrared (NIR) wavelengths. At SS1 in Algeria, the time

series of the scattering coefficients during the period of the campaign show dust events exceeding $3500 \ \mathrm{Mm^{-1}}$ and a relatively high mean SSA of 0.995 at 670 nm was observed at this site. The laboratory results show for the fine distributions in both sites a spectral dependence of the imaginary part of the refractive index Im(m) with a bow-like shape, with increased absorption in ultraviolet and also in the shortwave infrared. The same signature was not observed, however, in the mixed size distribution in Algeria. Im(m) was found to range from 0.011 to 0.001i for dust collected in Algeria and 0.008 to 0.002i for dust collected in

Mauritania over the wavelength range of 350-2500 nm. Differences in the mean elemental composition of the dust collected in the supersites in Algeria and in Mauritania and between fine and mixed modes distributions were observed from EDXRF measurements, although those differences cannot be used to explain the optical properties variability between the samples.





Finally, particles with low-density typically larger than 10 μm in diameter were found in some of the samples collected at the supersite in Mauritania, but these low-density particles were not observed in Algeria.

# 1   Introduction

Mineral dust originating from deserts and other arid surfaces is one of the most abundant aerosols in the atmosphere. According to Boucher et al. (2013), dust corresponds to 35% of the total continental aerosol mass of particles with diameter smaller than 10 μm. Roughly half of all aerosols above North America are dust particles that have been transported from other continents (Yu et al., 2012). Dust has a significant direct radiative effect on the Earth's energy balance, which globally acts to cool the planet. Depending on the fraction of the dust contributed by anthropogenic sources, the radiative forcing exerted on the climate system is estimated at -0.1 (-0.3 to +0.1) $Wm^{-2}$ (Boucher et al., 2013). To put that in perspective the total radiative forcing exerted by all aerosols is estimated to be -0.45 (-0.95 to +0.05) $Wm^{-2}$ (Boucher et al., 2013). Regionally, because aerosol forcing depends on the brightness of the underlying surface, over the Sahara itself dust imposes a positive radiative forcing primarily through longwave warming (Miller et al., 2014). Dust also plays a role in cloud microphysics, acting as ice nuclei and thereby influencing cloud development and subsequently ice cloud radiative effects and precipitation characteristics (Atkinson et al., 2013; Prenni et al., 2009). In addition to their effects on Earth's energy balance and water cycle, the transport of mineral dust particles are known to be important for biological productivity in ocean regions (Mahowald et al., 2009). Dust particles contain iron and phosphorous, and if these nutrients are bioavailable, when the dust is deposited into the ocean phytoplankton use these nutrients in photosynthetic activity (Jickells et al., 2005; Mahowald et al., 2008, 2009; Johnson and Meskhidze, 2013). Likewise, dust is known to bring important nutrients to the Amazon (Swap et al., 1992; Bristow et al., 2010; Rizzo et al., 2013; Yu et al., 2015). Long distance transport of dust contributes to air quality degradation (Yu et al., 2013; Prospero et al., 2014) and may be a means for intercontinental transport of biological and disease agents (Smith et al., 2012; Molesworth et al., 2002). The Sahara desert is the main source of dust, globally, contributing more than half of all global emissions, with an estimated amount of 182 million tons of dust carried across the western edge of the Sahara each year (Chin et al., 2009; Yu et al., 2015).

While we expect Saharan dust to affect Earth's climate system and biogeochemical cycles, quantifying the effect is still highly uncertain. Uncertainties are large because lacking strong observational constraints, diversity between model estimates of key aerosol properties and processes are large. For example, comparisons between different models show high variability in the prediction of the most straightforward aerosol property, total aerosol mass (Textor et al., 2006). This variability grows even higher when a specific type of aerosol is considered. For instance, for dust aerosols, models show a range in simulated atmospheric loading by a factor of four and a range of simulated emissions of nearly a factor of ten (Huneeus et al., 2011). A large part of this variability among various models predictions is associated with differences in the parameters used to describe emission, transport, and optical and microphysical properties of the aerosols (Textor et al., 2006). Observational constraints on Saharan dust are still too poor to bound estimates of the parameters necessary for quantitative determination of dust climate forcing and potential for fertilization of ecosystems. These parameters include dust emissions, lofting, transport, deposition,



composition, microphysical and optical properties. Specifically, while models have been constrained over the past fifteen years by global measures of aerosol optical thickness made by a constellation of satellite sensors (Lenoble et al., 2013), translating from the observed optical loading to a mass loading requires knowledge of the microphysical and optical properties of each individual aerosol type, and satellite sensors are incapable of providing this information.

The project, "Fennec – The Saharan Climate System" was conducted by a consortium of universities in France, U.K. and U.S.A in 2011 (Washington et al., 2012). This project joined efforts to address open questions on atmospheric processes in central Sahara. Combining aircraft (Ryder et al., 2013a, 2015), ground based (Marsham et al., 2013; Todd et al., 2013; Hobby et al., 2013), modeling and satellite observations (Banks et al., 2013; Chaboureau et al., 2016), the Fennec project successfully obtained a broad data set of meteorological conditions, atmospheric dynamics and structure, as well as dust

emission and transport mechanisms for the central Saharan region (Washington et al., 2012).

    The present study focuses on the ground-based measurements of the dust optical properties obtained using a custum-made Inverse Integrated Nephelometer and Optical Reflectometer developed by the Laboratory of Aerosol, Clouds and Optics (LACO) at University of Maryland, Baltimore County (UMBC) and subsequent detailed laboratory analyses of the samples collected by the LACO Aerosol Sampling Stations during the Fennec campaign. The *in situ* measurements were taken during the inten-

sive observation period, from end of May through June 2011. The LACO-UMBC instruments were deployed in two locations: Supersite 1 (SS1): Bordj Badji Mokhtar (BBM) in southern Algeria and in a small village called Bir Moghrein nearby the Fennec Supersite 2 (SS2) in Zourete, Mauritania. The instruments in both locations were operated by the Office National de la Meteorologie (ONM) of Algeria and Mauritania with remote assistance of the Fennec team.

    *In situ* measurements of Saharan dust were complemented with laboratory analyses for the characterization of their optical

properties. Size and aspect ratio distributions of the dust particles were obtained by scanning electron microscopy. Spectral optical reflectance measurements from the ultraviolet (UV) to the near infrared (NIR) wavelengths were obtained for each sample and the mean mass absorption efficiency and the imaginary part of the refractive index were derived for dust collected on filters at both supersites. The elemental composition of the dust samples was obtained by Energy Dispersive X-ray Fluorescence Analysis (EDXRF). Finally, our optical measurements were compared with a collocated AERONET sun-photometer in the

main supersite-1 in Algeria, when data were available.

    The next section places Fennec and the measurements presented here in context by providing a general background of previous campaigns and measurements of dust in and near the Western Sahara. Section 3 describes the sites where Fennec measurements were taken and the LACO-UMBC ground-based instruments deployed during the Fennec campaign. Section 4 presents the time series of the ground-based measurements and Section 5 describes the laboratory measurements of the sam-

ples collected during the campaign that allowed the derivation of the dust spectral mass absorption efficiency and imaginary refractive index. We intercompare our results with other measurements obtained during the Fennec and previous campaigns in Section 6. Finally, in Section 7, we present a discussion and the conclusions.





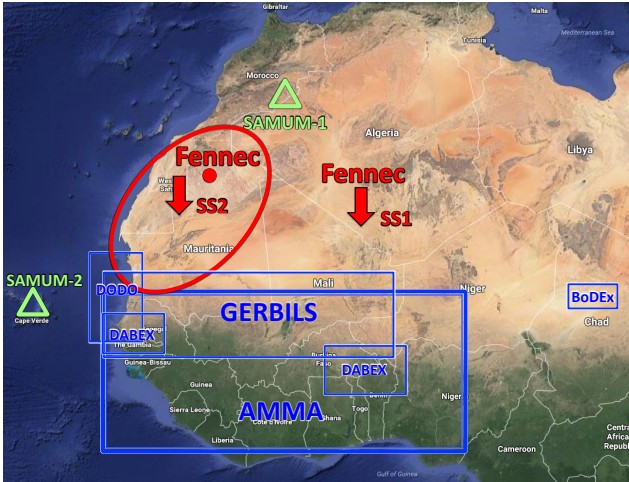

**Figure 1.** Northwestern Africa showing areas of operation of 3 major families of dust field campaigns. AMMA/DABEX/DODO/BoDEx is in blue boxes. SAMUM in green triangles. Fennec shown by red arrows pointing to the locations of Fennec Supersite 1 (SS1) in Bordj Badji Mokhtar in Algeria, Fennec Supersite 2 (SS2) in Mauritania at the cities of Zourete (main location). The red dot marks the city of Bir Moghrein, in Mauritania, where the second LACO Aerosol Sampling Station was deployed during the Fennec campaign.

## 2 Background

Project Fennec is one of a series of field campaigns deployed in and surrounding the Sahara desert engaged in characterizing Saharan dust. Focusing on the campaigns of the past dozen years, we group these into three families: (1) the Sahel and southern Sahara experiments of 2005-2007 (The Bodélé Dust Experiment – BoDEx, Dust and Biomass Experiment – DABEX, Dust

Outflow and Deposition to the Ocean – DODO, African Monsoon Multidisciplinary Analysis – AMMA, NASA AMMA – NAMMA, and Geostationary Earth Radiation Budget Intercomparison of Longwave and Shortwave radiation (GERBILS)) (Washington and Todd, 2005; Haywood et al., 2008; McConnell et al., 2008; Redelsperger et al., 2006; Zipser et al., 2009; Haywood et al., 2011), (2) the Moroccan and Cape Verde experiments of 2006 and 2008 (Saharan Mineral Dust Experiments – SAMUM1 and SAMUM2) (Heintzenberg et al., 2009; Ansmann et al., 2011), and (3) the Fennec climate programme of the

central and western Sahara of 2011 and 2012 (Marsham et al., 2013; Banks et al., 2013; Todd et al., 2013; Ryder et al., 2015). All three families included both a ground-based and airborne components. Figure 1 shows the general areas of operation of these three families of campaigns.

The AMMA/DABEX/DODO campaign was a broad investigation of the meteorology, aerosols and trace gases of the Sahel and southern Sahara (Haywood et al., 2008; McConnell et al., 2008). Ground sites, aircraft and modeling provided important

information on both mineral dust and biomass burning. These measurements clarified the microphysical distribution of these two aerosol types, their chemical composition and some information on microphysical and optical properties of the particles (Haywood et al., 2008; Chou et al., 2008; Osborne et al., 2008; Formenti et al., 2008; McConnell et al., 2010; Paris et al., 2010). Measurements in the southern Sahara were made during the dry season (northern winter) when both dust and biomass burning





aerosols are prevalent. The presence of biomass burning aerosols limited some characterization of pure dust, but sufficient pure dust cases were observed to determine dust aspect ratio, size distribution, extinction coefficient and single scattering albedo (at 550 nm), and compare these particle properties between locations in the southern Sahara to those near the Atlantic coast. The accumulation mode dust was found to be non-absorbing at 550 nm (Osborne et al., 2008) and the aspect ratio was 1.7

(Chou et al., 2008). Optical properties were estimated based on filter samples from DODO (McConnell et al., 2010) for short wavelengths only. Spectral optical properties were not measured.

The SAMUM campaigns targeted dust aerosol on the northwestern edges of the Sahara. SAMUM-1 in Morocco was chosen to be close to dust sources and relatively free from influence of biomass burning aerosols, and SAMUM-2 on the Cape Verde Islands was chosen to represent the dust and biomass burning outflow over the Atlantic (Ansmann et al., 2011). SAMUM

produced measurements for size dependent composition and aspect ratio. Unlike AMMA/DABEX, in SAMUM spectral optical properties were reported. Optical properties included spectral absorption coefficient, imaginary part of the refractive index and single scattering albedo (Ansmann et al., 2011; Kandler et al., 2009, 2011; Müller et al., 2009; Wagner et al., 2012). In some studies, complex refractive index was derived using mixing rules after mineral composition of the particles were determined (Kandler et al., 2009, 2011; Otto et al., 2009). In other studies a technique applied to aerosols collected on filters was used

to determine optical properties, including the imaginary part of the refractive index and single scattering albedo across a wavelength spectrum from 250 to 800 nm (Müller et al., 2009; Wagner et al., 2012).

In all of these campaigns differences in aerosol microphysical and optical properties were noted, dependent on mixtures of dust with other aerosol types and even for pure mineral dust. Differences were linked to locations: inland versus coastal (Osborne et al., 2008), Morocco versus Cape Verde (Ansmann et al., 2011), and northern versus southern fringes of the desert

(Ansmann et al., 2011). These differences were apparent even when using the same instruments and applying the same analysis techniques (Kandler et al., 2011), making clear that inherent differences exist in dust chemical, microphysical and optical properties. Fennec was designed to add new locations of dust sampling in the heart of the desert, including one site deep in the central Sahara (Fig. 1), and like previous campaigns approach dust characterization with a full array of ground-based, airborne and satellite observations and modeling (Marsham et al., 2013; Todd et al., 2013; Ryder et al., 2013a; Banks et al.,

2013). Fennec also built upon previous field campaigns with new technology and techniques that would aid in the overall characterization of the dust and its meteorological underpinnings, and in light of the present study, specifically in advances in the characterization of dust optical properties.

## 3 Instruments and sites

### 3.1 LACO Aerosol Sampling Station

UMBC-LACO deployed two automated LACO Aerosol Sampling Stations, one at each Fennec Supersite. The Aerosol Sampling Station is a system for collection of aerosol particles on filters designed and built at UMBC. This instrument has a cartridge with space for 16 filters, separated in 2 stages for 8 fine filters and 8 coarse filters. Nuclepore filters with 25 mm diameter and 5.0 and 0.4 $\mu$m pore diameters were used as coarse ($1^{st}$ stage) and fine ($2^{nd}$ stage) filters respectively to collect





the aerosol particles. Particles with aerodynamic diameters larger than 10 $\mu$m were removed by the aerodynamic impactor of the instrument inlet. This impactor has a cut efficiency of 50% for particles with aerodynamic sizes of 10 μm in dimeter and density equal 1 g/cm$^3$ (Hopke et al., 1997). For particles of density around 2.6 g/cm$^3$, such as dust, this is approximately equivalent a cut size of 50% at 6.1 $\mu$m geometric diameter.

The 1$^{st}$ stage filters adequately prevent coarse particles from passing through the pores to adhere to the 2$^{nd}$ stage filter. Thus the 2$^{nd}$ stage filter represents a fine mode aerosol and the size distributions analyzed from the 2$^{nd}$ stage filters include only particles with diameter less than 5 μm. The coarse particles in the sample adhere to the 1st stage filters, but so do many fine particles. There is overlap of size distributions of the 1$^{st}$ and 2$^{nd}$ stage filters, causing us to identify the 1$^{st}$ stage filter as representing a "mixed" size mode aerosol, rather than a coarse mode.

Each sampling position in the cartridge is connected individually through vacuum tubes to the control system unit containing automatic valves, flow meters, pump controller, and the data acquisition system. The filters were pre-weighed and the cartridges were prepared, individually labeled, and packed at the LACO filter laboratory at UMBC to avoid in field contamination. The filter in the eighth position of each cartridge was not sampled and was used as reference blank. The airflow pumped by the sampling station through the filters was set at 4 liters per minute (LPM). At the end of the campaign, cartridges containing

the sampled filters were sent back to the laboratory at UMBC for detailed analysis of mass, size and aspect ratio distribution, chemical composition and spectral optical reflectance measurements. See Table 1 for deployment durations and sampling periods.

### 3.2  Inverse Integrating Nephelometer and Optical Reflectometer

The Inverse Integrating Nephelometer and Optical Reflectometer (N-OR system) were designed to make real time measure-

ments of the scattering and absorption coefficients of ambient aerosol particles. This instrument connects an Inverse Integrated Nephelometer (N system) with an Optical Reflectometer (OR system) into a single unit that was designed, built, and tested at the LACO at UMBC.

The N system component measures the total scattering coefficient integrated over an angular range of 5-178 degrees. An aerodynamic impactor, in the inlet of the equipment cuts off particles larger than 10 μm in diameter. The internal laser beam

with wavelength of 670 nm illuminates particles entering the inlet of the instrument. A photomultiplier tube (PMT) detector and a cosine diffuser are positioned perpendicular to the laser beam aiming to maximize the scattering angle coverage of the instrument.

At the end of the N system, the OR system component measures the change of reflectance of a Nuclepore filter in real time as the particles collect on the filter and darken the surface. The OR system uses three LEDs at wavelengths 450, 530 and

640 nm, to illuminate the filter consecutively and allow for derivation of absorption at these three wavelengths. This allows the simultaneous measurement of the scattering and absorption coefficients of the aerosol particles, the calculation of single scattering albedo for the 670 nm in real time (by scaling the absorption coefficient from 640 nm to 670 nm using our spectral measurements discussed on Sec. 4.4), and the creation of time series of these optical parameters with a temporal resolution of 45 seconds.





**Table 1.** Characteristics and sampling period of the LACO-UMBC instruments deployed during the Fennec campaign.

| Location | Instrument | Measurement | Sampling period | Deployment duration |
|---|---|---|---|---|
| Algeria | LACO Aerosol Sampling Station | Filter samples | 1.5-12 h | 6-30 June |
| | Real-time Optical Reflectometer | Optical reflectance | 45 sec | 22-30 June |
| | Inverse Integrating Nephelometer | Scattering coefficient | 4 sec | 6-30 June |
| Mauritania | LACO aerosol sampling station | Filter samples | 6 h | 23 May - 26 June |

Table 1 summarizes the characteristics and sampling periods of the ground-based measurements obtained by the LACO-UMBC instruments during Fennec 2011.

### 3.3 Fennec supersites

The LACO-UMBC instruments were deployed at Supersite 1 (SS1): Bordj Badji Mokhtar (BBM) in the heart of the central Sahara, and in a small village called Bir Moghrein approximately 290 km north of Supersite 2 (SS2) that was located in the city of Zourete, Mauritania. The reason for the deployment at a distance from SS2 was to avoid contamination from aerosols produced from local mining operations in Zourete. See Fig. 1.

Supersite 1 (SS1) is located in BBM–Algeria (21.38 N, 0.92 E, ≈420 m above sea level). In addition to the LACO-UMBC instruments, the Fennec team deployed other instruments at this location, including an AERONET Cimel Sun photometer. A detailed description of the other instruments and measurements of the Fennec campaign at SS1 is available in Marsham et al. (2013). Figure 2 shows an image of the tower where the instruments were installed. The inlets of both aerosol sampling stations and the Nephelometer were positioned at a distance of 3 meters from the ground. The installation of this tower close to the ONM and to the airport facilitated access to the tower for replacement of filter cartridges. Also, it allowed the operation of the aerosol sampling station in manual mode, in which the operator collected more filters during intense episodes of dust.

The location of the second automated LACO Aerosol Sampling Station was in the remote town of Bir Moghrein (location 25.23 N, 11.62 W, ≈360 m above sea level). The aerosol sampling station was operational from 23 May to 26 June 2011 collecting three to four filters per day with its inlet also located 3 meters above ground. Given the difficulty in accessing the Bir Moghrein site, the station was preset to automatically sample filters in the following periods of time: 07:00 to 13:00 UTC, 13:00 to 19:00 UTC, 19:00 to 21:00 UTC, and 21:00 to 07:00 UTC. No Nephelometer-Optical Reflectometer was deployed at Bir Moghrein, and therefore no high temporal resolution data was collected with the LACO-UMBC instruments at this site.



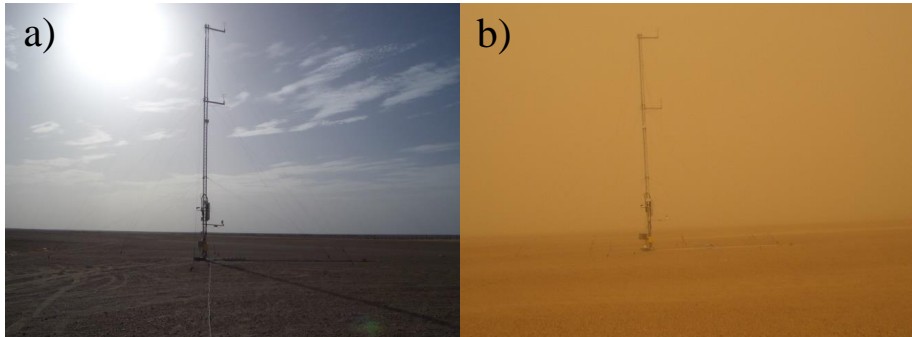

**Figure 2.** Tower at SS1 in Bordj Badji Mokhtar in Algeria with the LACO-UMBC instruments during an episode of low (left) and high (right) concentration of dust aerosol taken on June, 8 and 17, respectively. The Inverse Integrating Nephlometer, Optical Reflectometer and, the LACO aerosol sampling station were installed with inlets three meters above the ground level. Image credits: Mohammed Salah and Bouzine Ouchene – ONM Algeria.

## 4 Time Series of Dust Characterization

### 4.1 Time series of mass concentration

LACO Aerosol Sampling Stations were deployed at both sites, allowing for measurements of aerosol mass concentration as function of time, with a resolution of 6 hours, except during intense dust episodes at SS1 where samples were collected
at a higher frequency. The sampling station automatically advanced measurements from filter to filter during measurement periods defined in Section 3.2, accumulating aerosol mass on four different filters each day. After the end of the campaign, cartridges containing the sampled filters were sent back to UMBC. At the laboratory, each filter was post-weighed and the mass collected in each filter was obtained. The mean mass aerosol concentration for the period that each filter was sampled was obtained by dividing the sampled mass by the integrated flow of the sampling period of each filter. The temporal resolution
of the mass concentration time series is nominally six hours, based on the six hour sampling period of each filter, and the mass concentration time series is not a real-time measurement. The sampling station filter cartridge supports two size-stages for each sampling period, as defined in Section 2.2, and thus the mass concentration time series is available for both the aerosol loading on the coarse pore ($1^{st}$ stage) and the fine pore ($2^{nd}$ stage) filters.

Figure 3 shows the concentration (in $\mu g/m^3$) from the $1^{st}$ (mixed) and $2^{nd}$ (fine) stage filters for both stations in (a) Algeria
and (b) Mauritania. Note that while the 1st stage filter successfully prevents coarse particles (d > 5 μm) from passing through to the $2^{nd}$ stage, allowing the 2nd stage filter to represent a true fine mode aerosol, both fine and coarse particles adhere to this first stage, creating a mode of mixed sized particles. This size separation of the two filters will be shown in Section 5. In Fig. 3, we see that the mass concentration at SS1 in Algeria reached levels approximately ten times larger than in Mauritania. In Algeria, the highest peaks of mass concentration were observed on 13 and 18 of June, with lesser events noted on 16, 21-22
and 29-30 of June. These peaks are associated with the sudden moistening convective events described by Marsham et al.



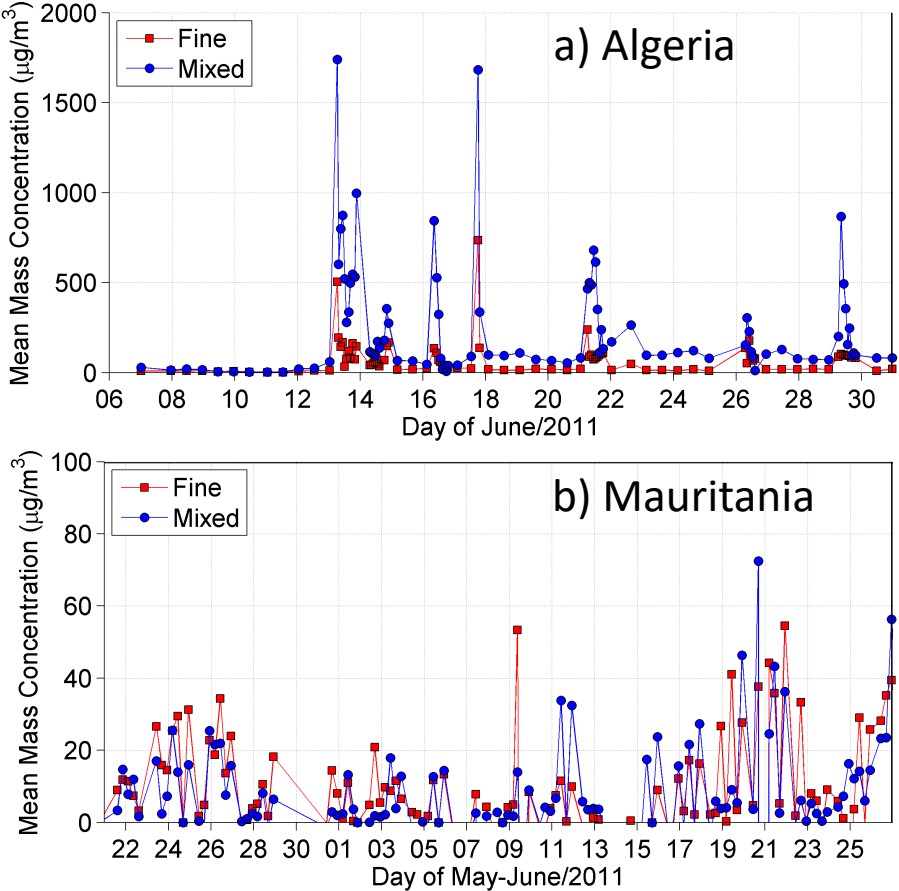

**Figure 3.** Mass concentration of dust in $\mu g\,m^{-3}$ collected on filters using the LACO aerosol sampling stations for both supersites in (a) Algeria and (b) Mauritania. Each data point represents the average mass concentration for the given sampling period. Fine mode mass concentration is calculated from the $2^{nd}$ stage filters. Mixed mass concentration is calculated from the $1^{st}$ stage filters where both fine and coarse particles adhere to the surface. Note the different scales on the y-axis in plots (a) and (b). Uncertainties were estimated to vary between 3.0 and 7.0 $\mu g\,m^{-3}$ for days with low and high mass concentrations respectively.

(2013). The $25^{th}$ is also a moistening event, but does not have a corresponding peak in mass concentration as measured by the Aerosol Sampling Station at SS1. In Mauritania, we see the distinction between the "maritime phase" with low aerosol loading that occurs from 1 to 13 June and the onset of the "heat low phase" after that period with higher aerosol loading and greater influence from the interior desert, as described by Todd et al. (2013).

## 4.2 Time series of aerosol scattering coefficients

Unlike the mass concentration time series that is derived post-deployment by analyzing the series of individual filters from each Aerosol Sampling Station, yielding a time series with temporal resolution of approximately 6 hours at both SS1 and SS2,




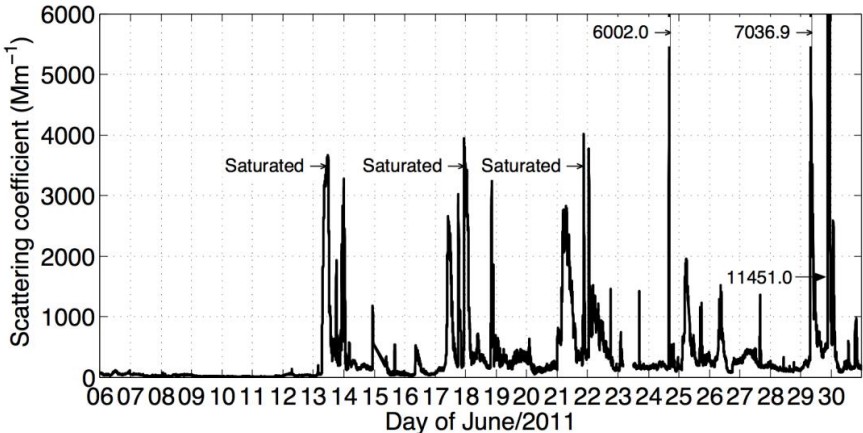

**Figure 4.** Integrated Scattering Coefficient in $\mathrm{Mm}^{-1}$ measured at 670 nm during the Fennec campaign at SS1 in Algeria in the period of June 6-30, 2011. Before June 22, events of dust that had its scattering coefficient exceeding $3500\,\mathrm{Mm}^{-1}$ saturated the equipment, as marked in the plot. Uncertainties were estimated to be within 5%.

the time series of integrated scattering coefficient ($\beta_{sca}$) was measured in real time every four seconds using the N system located only at SS1. Figure 4 shows the integrated scattering coefficient (in $\mathrm{Mm}^{-1}$) for the whole period of the campaign. In the first week, there was a long period with relatively low dust loading followed by a sequence of intense episodes of high dust concentration. On June 13, the high concentrations of aerosol exceed the saturation limits of the nephelometer, and all the data above $3500\,\mathrm{Mm}^{-1}$ were not measured. On June 22, the detection scale of the nephelometer was reconfigured in order to allow for higher dynamic range and prevent saturation. The period from June 22 to June 30 has the optimum configuration conditions for the N-OR system. Note that the days of peak scattering coefficient at SS1 correspond to some of the same days of independently measured high mass concentration, i.e. 13, 18, 21-22 and 29-30 June. The $25^{th}$ of June also shows a high scattering coefficient, but that day is missing from the mass concentration time series, although, the $25^{th}$, like the other observed high mass/high scattering events, follows a moistening event identified in Marsham et al. (2013).

## 4.3 Time series of aerosol absorption coefficient and single scattering albedo

Simultaneous to the scattering coefficient measurements, the same dust particles that passed through the nephelometer are collected on filters and measured in real time by the reflectometer. Due to a technical problem, the reflectometer did not work properly at the beginning of the campaign. The reflectometer data presented in the next sections were obtained after June 22, after the problem had been identified and fixed.





The mass absorption efficiency ($\alpha_{abs}$), in $\mathrm{m^2\,g^{-1}}$ is the parameter derived from the measured quantities of attenuated reflectance of the filter ($I/I_0$) and aerosol mass concentration ($\sigma$), in $\mathrm{g\,m^{-2}}$. The relationship between the measured quantities and $\alpha_{abs}$ is given by Eq. (1),

$$\sigma = \frac{G}{2\alpha_{abs}}\left[-\ln\left(\frac{I}{I_0}\right)\right]^b,\qquad(1)$$

where $I$ is the measured reflectance and $I_0$ is the reflectance of the clean pristine filter. The functional relationship between $\sigma$ and the reflectance ratio is a power law of the logarithm, where $b$ is an empirical power law coefficient determined to be 1.218 and $G$ is a geometrical factor determined to be equal 1 for a large range of geometries, including the one used in this work. This method was previously derived and calibrated by Martins et al. (2009) using Monarch 71 black carbon particles

manufactured by the Cabot Corporation, it was compared and showed good agreement against other absorption techniques by Reid et al. (1998) and applied to volcanic ash samples (Rocha-Lima et al., 2014). This technique is based on the assumptions that the reflectance of the Nuclepore filter cannot be increased and that the reduction of the light reflected by the filter is due to only to absorption by the aerosol particles.

The absorption coefficient ($\beta_{abs}$) in $\mathrm{Mm^{-1}}$ is $\alpha_{abs}/\rho$, where $\rho$ is the aerosol concentration ($\mathrm{g\,m^{-3}}$). For real time calculations

of $\beta_{abs}$ using the reflectometer we measure the darkening of the filter as a ratio of reflectance at two points in time, $t_1$ and $t_2$, with $I(t_1)$ substituting for $I_0$ of Eq. (1). The darkening of the filter is thus relative to the previous measurement and is no longer referenced to the pristine filter. Then, the power law equation given by Eq.1 was rewritten in terms of the linear absorption coefficient $\beta_{abs} = \alpha_{abs}/\rho$, where the concentration $\rho = M/V$ (g/m³) depends on the aerosol mass $M$ collected and the volume $V$ of air that passed through the filter in the time interval $\Delta t = t_2 - t_1$. The total volume is obtained by integrating

the measured flow $F$ in time, i.e., $V = F \cdot \Delta t$. Using that the aerosol mass concentration $\sigma$ as the ratio of the aerosol mass $M$ and the sampled area of the filter $A_{filter}$, in the Eq. 1, the dependence with the aerosol mass $M$ cancels out.

$$\beta_{abs} = \frac{1}{2}\left[-\ln\left(\frac{I(t_2)}{I(t_1)}\right)\right]^{1.218} \cdot \frac{A_{filter}}{F \cdot \Delta t} \cdot 10^6 \qquad(2)$$

Figure 5 shows the results of the simultaneous measurements of scattering and absorption coefficients, ($\beta_{sca}$ and $\beta_{abs}$), at

SS1 for the period of June 22-30. Figure 5 (a) shows the scattering coefficient in $\mathrm{Mm^{-1}}$ for 640 nm. This is a subset of the plot in Fig. 4. During this period several episodes of high concentration of dust were detected. The largest episodes of dust were seen on 24, 29 and 30 day of June. Scattering measurements were taken every 4 seconds and are shown in the plot averaged every 45 seconds in order to correspond to the time scale of the reflectometer. Figure 5 (b) shows the reflectance of Nuclepore filters at three wavelengths normalized by the reflectance of the clean filter measured at the beginning of the sampling. Filters

in the Reflectometer were replaced approximately once per day. The sampling start time of each filter can be identified as the





moment where the reflectance is close to 1. The slope of the curve of the reflectance is proportional to the amount of aerosol in the filter at that moment and therefore to the concentration of the particles collected on the filter at that given instant.

Figures 5 a) and c) show similar trend between scattering and absorption coefficients, obtained by the nephelometer and the reflectometer, respectively. The uncertainties in the scattering measurements were estimated to be smaller than 5% for highly scattering particles such as dust particles. The uncertainties in the absorption coefficient were estimated from the error in the reflectance, integrated flow, and size of the filter where particles were collected in the filter. These errors combined represent an uncertainty on the order of 3% (or $2\,\mathrm{Mm}^{-1}$, whichever is higher) in the absorption coefficient.

The single scattering albedo (SSA) is defined as $\beta_{sca}/(\beta_{sca}+\beta_{abs})$. By obtaining simultaneous scattering and absorption coefficients at SS1, the calculation of a time series of SSA was possible at 670 nm, the wavelength measured by the nephelometer. Because $\beta_{abs}$ is obtained at 640 nm, this value of was extrapolated to 670 nm based on spectral laboratory measurements showed in Fig. 8 (a). The SSA time series are shown in Fig. 5 (d). These results show variation along this period from 0.96 to close to 1, with a mean value around 0.995, which is around 0.01 to 0.02 systematically higher than the values for Saharan dust found in the literature at this wavelength.

## 5 Spectral imaginary part of the refractive index of dust

In addition to the time-resolved measurements of aerosol mass and optical properties, further analysis of the dust-laden filters obtained during Fennec can reveal time-integrated properties of the dust particles, namely the spectral imaginary part of the refractive index. Using the same methodology applied in Rocha-Lima et al. (2014), the derivation of the imaginary refractive index was obtained by minimizing the difference between the mean mass absorption efficiency derived from direct measurements of the attenuated reflectance on the filters (Eq. 2) and that calculated from Mie or T-matrix theory using size and shape parameters obtained from other analysis of the particles on the filters. The independent calculation of mass absorption efficiency is governed by,

$$\alpha_{abs} = \frac{\int_0^\infty n_A(r)Q_{abs}(\mathrm{m},x)A(r)\mathrm{d}r}{\int_0^\infty n_A(r)d_p v(r)\mathrm{d}r}, \tag{3}$$

where $n_A(r)dr$ is the number of particles per unit of area with radii in the range $[r, r+dr]$ in a given microscopic area, dp is the grain density of the particles, and $v(r)$ is the volume of each particle. The absorption efficiency $Q_{abs}(\mathrm{m},x)$ is a function of the complex refractive index $(m)$ and size parameter $(x)$, and was obtained by applying either Mie or T-Matrix theory following the same method applied in Rocha-Lima et al. (2014). For all calculations the real part of the refractive index was held constant spectrally at a value of 1.56 (Balkanski et al., 2007; Petzold et al., 2009). In a similar method, Wagner et al. (2012) fixed the real part of the refractive index to be 1.53 to derive the imaginary part of the refractive index. The imaginary part of the refractive index that yields the $Q_{abs}(\mathrm{m},x)$ producing the closest calculated $\alpha_{abs}$ to the measured value of $\alpha_{abs}$ from Eq. 2 is identified as the retrieved value. The retrieval is performed for the entire range of wavelengths from 350 to 2500 nm. This derivation requires laboratory measurements of the spectral optical reflectance of the filters using a spectrometer to obtain $\alpha_{abs}$ from





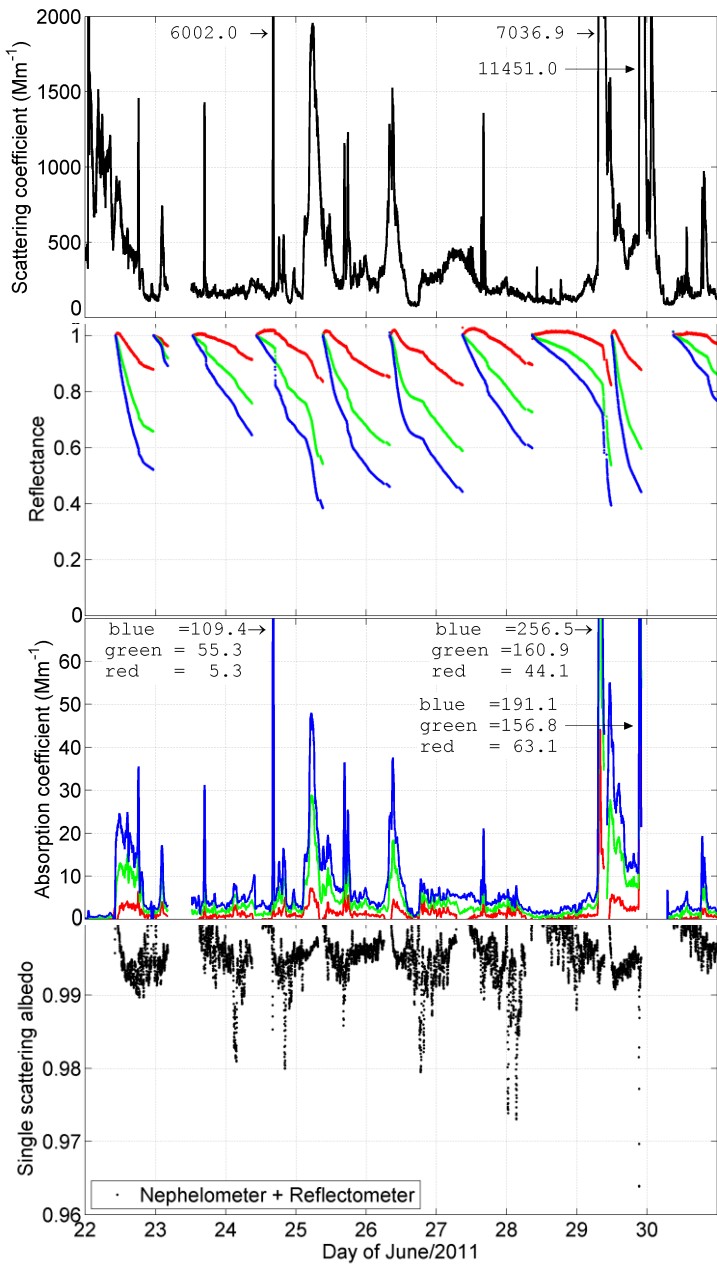

**Figure 5.** *In situ* measurements from SS1 in Algeria a) scattering coefficient in $\mathrm{Mm}^{-1}$, b) reflectance measurements normalized with respect to a clean filter at three wavelengths, 450 nm (in blue), 530 nm (in green), and 670 nm (in red), c) derived absorption coefficient for the period of June 22 to June 30, 2011, and d) Single scattering albedo of the Saharan dust at 640 nm calculated by combining the measurements of scattering and absorption coefficients from the nephelometer and the reflectometer, respectively. Uncertainties are discussed in Sec. 4.3.




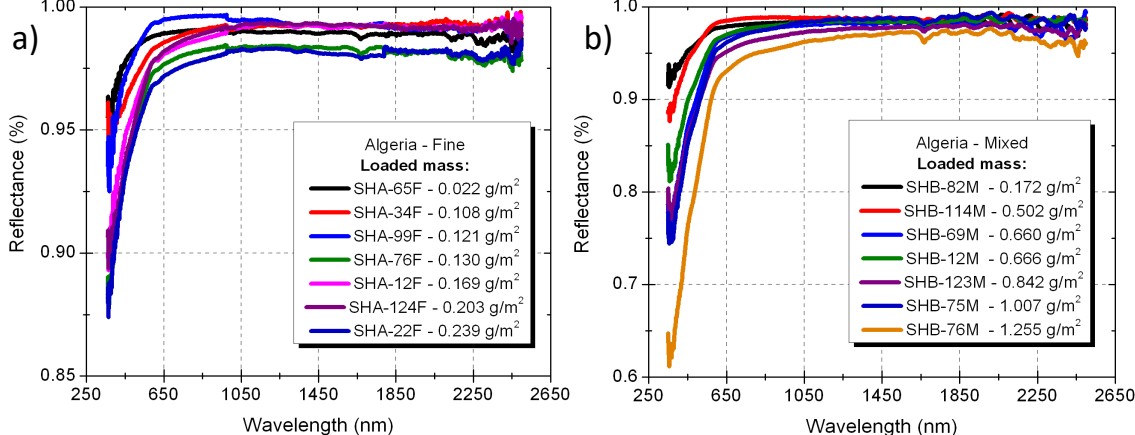

**Figure 6.** Examples of typical spectral reflectances of the Sahara dust at SS1 in Algeria for a) fine mode ($2^{nd}$ stage filter) and b) mixed (fine+course) mode ($1^{st}$ stage filter), according to the loaded mass per unit area $\sigma$ of each filter in $[\mathrm{g\,m^{-2}}]$. Each curve represents the average over 25 measurements of reflectance over the same filter. Uncertainties on the reflectance were estimated to be a maximum of 2.0% for the full wavelength range.

Eq. 2, measurements using a Scanning Electron Microscope (SEM) to obtain size distribution and aspect ratio of the particles, a calculation of particle density and radiative modeling of the particles using either a Mie or T-matrix code, as described in the following subsections.

### 5.1 Spectral optical reflectance measurements and derivation of mass absorption efficiency

The spectral reflectance from 350 to 2500 nm was obtained for all sampled filters relative to blank filters using a FieldSpec Pro from Analytical Spectral Device in the wavelength range of 350 to 2500 nm and a reflectance lamp from ASD Inc. The method applied in this analysis followed the same experimental procedure used in Rocha-Lima et al. (2014).

The reflectance of the filters collected at both stations was measured relative to a white reference. Figure 6 shows examples of reflectance spectra for fine and mixed modes for filters of different mass loading collected in Algeria (SS1). The typical

reflectance spectrum obtained for the filters in Algeria shows a sharp decrease in reflectance for wavelengths less than 650 nm. This strong spectral dependence is what causes the dust to appear brown to our eye. Some of the samples also presented a slight decrease in reflectance above 1000 nm. The ripples on the reflectance curve above 2000 nm are measurement artifacts also observed in clean filters.

The reflectance of the filters collected in Mauritania presented significant spectral variations and three groups of samples were identified, as shown in Fig. 7. Group 1 shows spectral reflectance similar to the reflectance of the samples collected

in Algeria as shown in Fig. 6, with strongly decreased reflectance below 600 nm. Filters from the Group 2 have reflectance spectra with a flat signal spanning the UV to visible wavelengths, a minimum of reflectance around 860 nm, and then a slightly





increasing reflectance as a function of wavelength through the shortwave infrared. Finally, Group 3 has a reflectance spectrum that seems to be a combination of Groups 1 and 2.

Scanning Electron Microscopy (SEM) images of these three groups show that Groups 2 and 3 contain extra-large particles comparatively to the sizes from Group 1. While the SEM images show irregularly shaped particles, reminiscent of dust, our laboratory observations indicated that these extra-large particles have lower density compared with typical dust. Firstly, the mass of the filters from Group 2 is relatively low, even though SEM images show a considerable number of particles on them. Secondly, these large particles are easily damaged by the electron beam of the microscope. Finally, the presence of the aerodynamic impactor with nominal cutoff size of 10 μm (or approximately 6.1 μm for a typical dust particle of density 2.6 $\mathrm{g\,cm^3}$) in the inlet should have removed most of these particles, unless they have lower inertia and lower aerodynamic sizes, i.e., lower mass density.

Thus, while the spectral reflectance, size and density of the particles of Mauritania Group 1 resembles the dust properties and the measurements from SS1, the measured particle properties of Mauritania Groups 2 and 3 do not. The anomalous spectral reflectance and particle sizes of Groups 2 and 3 are always linked together, meaning we do not find filters with Group 2 or 3 spectral reflectance without also finding Group 2 and 3 particle size and density properties. These particles are sparse on the surface of the filters and the complete characterization of their properties and origin would require dedicated microscopy and trajectory analysis, which are beyond the scope of this work. Therefore, for the remainder of the analysis we will focus on the properties measured and derived from Algeria and from Mauritania Group 1 only.

From the spectral reflectance measurements and mass concentration applied to Eq. 2, we derived the spectral mass absorption efficiency (in $\mathrm{m^2\,g^{-1}}$) for fine and mixed distributions for SS1 and SS2 (Group 1 only), as shown in Fig. 8. The fine and mixed size distributions correspond to the particles on the $2^{nd}$ and $1^{st}$ stage filters, respectively, as discussed in Section 4.2. For SS1, the mass absorption efficiency of both fine and mixed modes is in good agreement up to wavelength of 600 nm. Above that, fine and mixed mode deviate from each other, with the fine mode exhibiting higher values. For SS2, the mass absorption efficiency for the fine mode is slightly higher than the mixed mode for wavelengths up to 600 nm, and both modes are compatible above that. The uncertainties of these curves are represented by the bands plotted around the central value of the mass absorption efficiency corresponding to one standard deviation.

## 5.2 Size distribution measurements

The number, area, and volume size distributions were obtained from SEM images of the dust particles. Analysis included both the $1^{st}$ stage filters with pore size 0.4 μm and the $2^{nd}$ stage filters with pore size 5 μm as shown in Section 3.2.

Figure 9 a) and b) show the fine and mixed distributions for Algeria and Mauritania (Group 1), respectively, obtained by analyzing approximately 2000 particles. In this example, these distributions shows considerably larger fraction of particles below 1um of diameter in Mauritania, compared with Algeria. Other studies also show a decrease in coarse mode fraction as sampling moves towards aged dust and away from fresh dust near sources (Weinzierl et al., 2009, 2011; Ansmann et al., 2011; Ryder et al., 2013a, b).





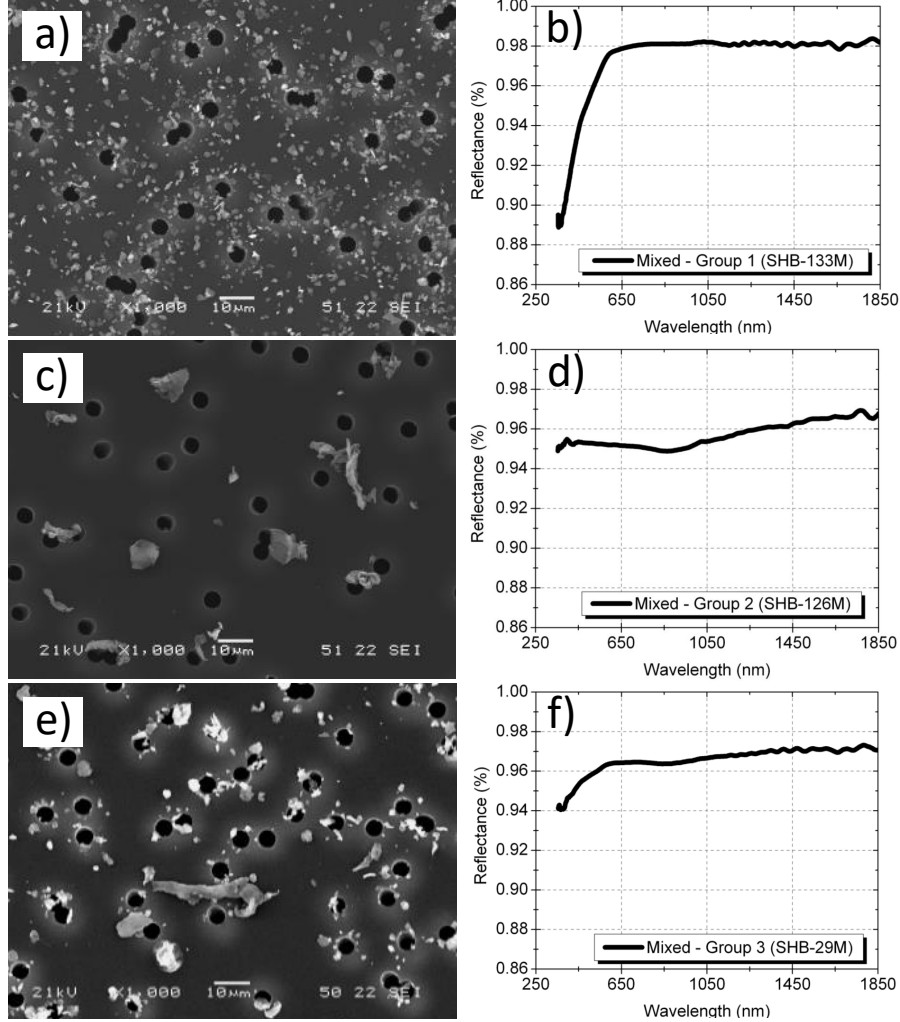

**Figure 7.** Examples of spectral reflectance and SEM images of Saharan dust from SS2 in Mauritania for a) and b) Group 1, c) and d) Group 2, and e) and f) Group 3. Each curve represents the average over 25 measurements of reflectance over the same filter. Uncertainties on the reflectance were estimated to be of a maximum of 2.0% for the full wavelength range. The scale bar in the SEM images shows a 10 μm scale for size reference. The samples were collected respectively, on June 21, June 19, and May 26, 2011.

The geometric distribution obtained by SEM images is the distribution sampled on the filter and for consistency this is the size distribution used to derive the optical properties of the dust collected on the filters using post-deployment measurements in the laboratory. A comparison between this geometrical distribution obtained by SEM and other size distributions retrieved from optical measurements or using different aerodynamic inlets must take into account the impactor efficiency for aerodynamic

5   sizes below 10 μm.




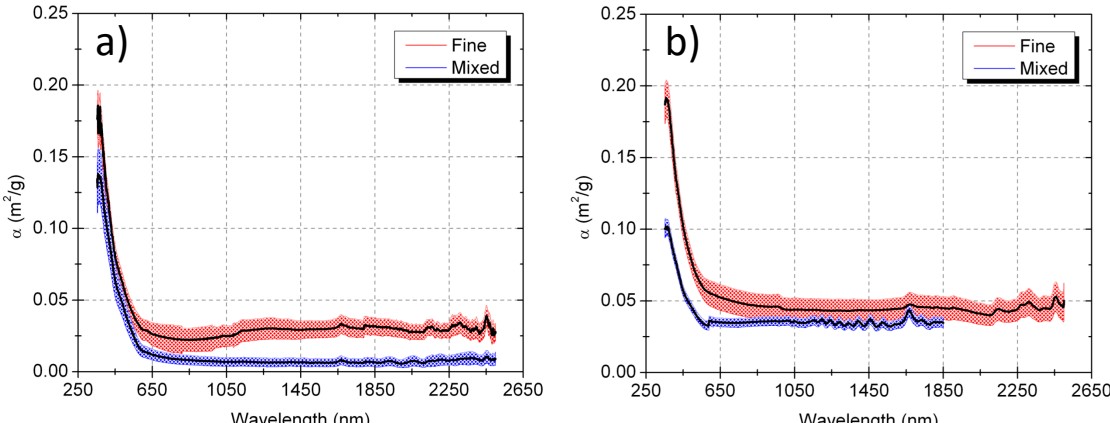

**Figure 8.** Spectral mass absorption efficiency ($\alpha_{abs}$) for fine and mixed mode particles of the Saharan dust collected on filters during the Fennec campaign a) Algeria and b) Mauritania (Group 1). Uncertainties shown as error bands in this figure were estimated by propagating the error from the power law fitting, and they represent one standard deviations around the central black lines.

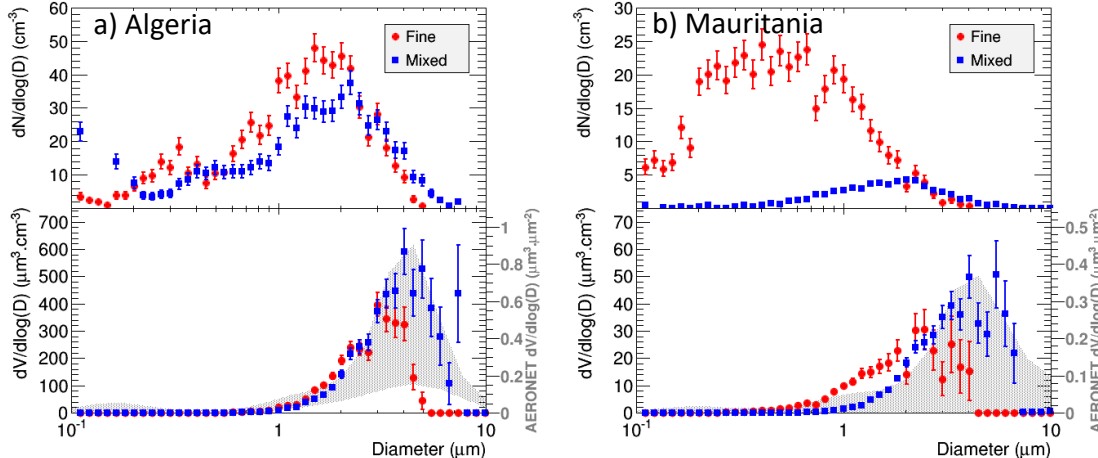

**Figure 9.** Particle number and volume distribution versus particle diameter obtained by analysis of SEM images for a fine and a coarse filter of Saharan dust sampled in a) Algeria and b) Mauritania (Group 1). The size distribution obtained by SEM corresponds to the geometrical size of the particles. In the lower panel in gray is shown the AERONET size distribution for the period of the campaign for a) the BBM site, collocated with the LACO-UMBC Aerosol Sampling Station and b) the Zourete site, approximately 290 km from Bir Moghrein. Note that AERONET volume density is per unit area, not volume, and is thus plotted with its own y-axis scale, shown in gray on the right hand side of the figures.





### 5.3 Determination of grain density

The technique used for the measurements of the grain density is based on the determination of the volume of the sample using the principle of gas displacement in a device under compressions and it requires a bulk sample of at least 1–2 grams of the material, as was described in Rocha-Lima et al. (2014). The major dust storms in Algeria caused significant uplift and deposition of dust on the surface of the instruments. Once the instruments arrived back at UMBC, dust deposited on the instrument surfaces was gently collected using a brush and sieved using a 45 µm mesh grid. The resulting bulk sample obtained had the required mass needed for the grain density measurements. The average grain density obtained for the Saharan dust from Algeria was $2.69\pm0.12$ g cm$^{-3}$. Because there was not enough material from SS2 in Mauritania for a grain density analysis, we used the same grain density for the samples of the Group 1 collected at the supersite in Mauritania.

The grain density for dust particles reported in the literature range from 2.1–2.6 g cm$^{-3}$ (Chen et al., 2011; Reid and Maring, 2003; Reid et al., 2008; Wagner et al., 2012). Ryder et al. (2013a) used 2.65 g cm$^{-3}$ to parameterize dust density during Fennec's airborne measurements and these values are compatible with our values measured in the laboratory.

### 5.4 Determination of particle aspect ratio

Now that size distribution and particle grain density have been determined, the final input needed to calculate $\alpha_{abs}$ from Eq. (3) is $Q_{abs}(\mathrm{m}, x)$. If we assume that the particles are spherical, we can use a Mie code to calculate $Q_{abs}(\mathrm{m}, x)$. However, dust particles are not always spherical, so that in addition to the Mie code, the extended-precision T-matrix code (Mishchenko et al., 1996) assuming randomly oriented ellipsoidal particles was used with a modified gamma distribution fitted to the measurements. The T-matrix code requires aspect ratio of the particles as input. For the fine mode, the value of the most probable aspect ratio used was obtained as 1.3 for both supersites from the analysis of SEM images, taken from the distribution shown in Fig. 10. This is for the fine mode only, as the T-matrix code does not converge for coarse particles in the wavelength range we are studying. We note that aspect ratio of mineral dust was measured to be 1.7 during AMMA/DABEX (Chou et al., 2008; Haywood et al., 2008), and 1.7–1.9 from samples collected during SAMUM (Wagner et al., 2012). In Morocco for dust sizes with diameter $>$ 0.5 µm aspect ratio was 1.6 (Kandler et al., 2009). In Cape Verde, similar aspect ratios were found (Kandler et al., 2011).

However, in a laboratory analysis of size separated mineral dust the aspect ratio was 1.3 for small dust particles measured during SAMUM (Ansmann et al., 2011; Kandler et al., 2009), similar to the results of our SEM analysis of the fine mode filters.

### 5.5 Derivation of spectral imaginary part of the refractive index

Finally, the imaginary part of the refractive index of the dust particles was derived using a minimization method applied for the mass absorption efficiency for each wavelength and the results are shown in Fig. 11. This minimization consists of finding the imaginary part of the refractive index in which the mass absorption efficiency derived from measurements of optical reflectance (Fig. 8) matches the mass absorption efficiency calculated using Eq. 3. The real part of the refractive index

(c) Author(s) 2017. CC BY 3.0 License.





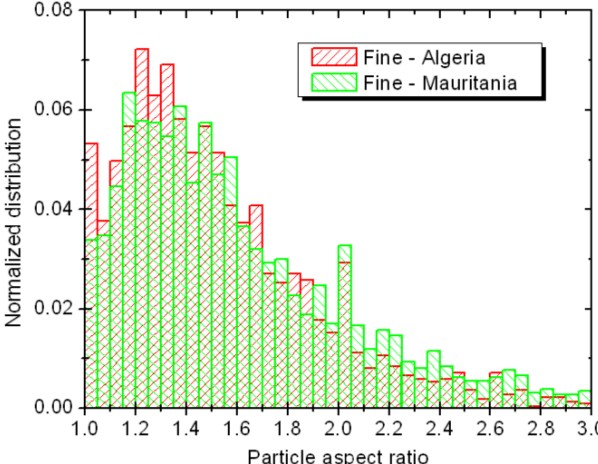

**Figure 10.** Particle aspect ratio distribution obtained using Scanning Electron Microscopy (SEM) analysis for the fine mode distribution of Saharan dust from Algeria and from Group 1 in Mauritania.

is assumed to be a constant value of 1.56 for all wavelengths. In the calculation of the absorption efficiency $Q_{abs}(m, x)$, the shape of the fine particles was considered to be first spherical and then spheroidal.

Figure 11 a) shows that the imaginary part of the complex refractive index for Saharan dust from Algeria has significant spectral differences between fine and mixed mode. Both fine and mixed modes present a significant increase inversely pro-

portional to wavelength below 600 nm. For longer wavelength the values diverge considerably, as the imaginary part of the refractive index of the mixed fraction remains nearly constant relative to the values found for the fine fraction, which increases significantly as a function of the wavelength. Similarly, the same behavior found for the Algeria fine mode is observed for both fine and mixed mode in Mauritania. For the mixed mode in Mauritania, the mass absorption efficiency and refractive index were derived for wavelengths up to 1850 nm. Above this wavelength the minimization method did not converge within an

acceptable error of 5%.

In a similar analysis Wagner et al. (2012) derive the spectral imaginary part of the refractive index for a variety of mineral dust samples, including samples collected during SAMUM in Morocco. We compare our retrievals with their results in Section 7.

**5.6   EDXRF analysis of Saharan dust**

Selected dust samples collected in the Sahara were submitted to Energy Dispersive X-ray Fluorescence analysis (EDXRF)

using an Epsilon 5 PanAnalytical spectrometer at the Atmospheric Physics Laboratory at University of Sao Paulo. A total of 150 samples, including $1^{st}$ and $2^{nd}$ stage filters from both supersites, were randomly selected for this analysis. Figure 12 shows the average concentration in percentage of the total mass of the main elements measured for samples from Algeria and Mauritania.




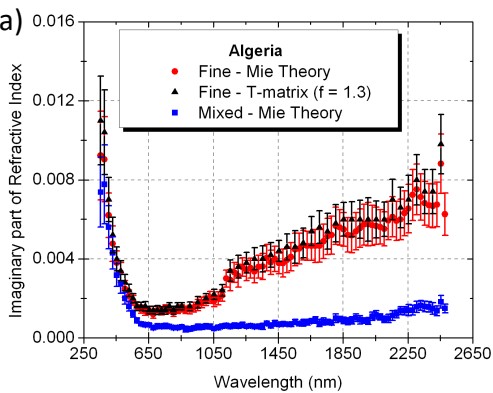
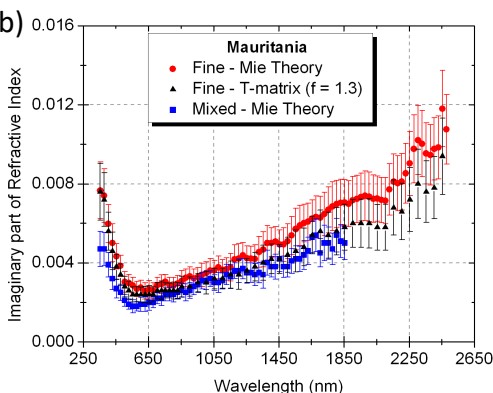

**Figure 11.** Imaginary part of the complex refractive index derived for mixed and fine particles of Saharan dust from a) Algeria and b) Mauritania (Group 1). Mie theory and T-matrix was used assuming the real part of the refractive index Re(m) = 1.56 and grain density $d_p = 2.69\,\mathrm{g\,cm^{-3}}$. The error bars of the imaginary part of the complex refractive index were estimated by studying the sensitivity of the minimization method to the uncertainties of the real part of the refractive index, the mass absorption efficiency, the particles' cross section, volume, and grain density.

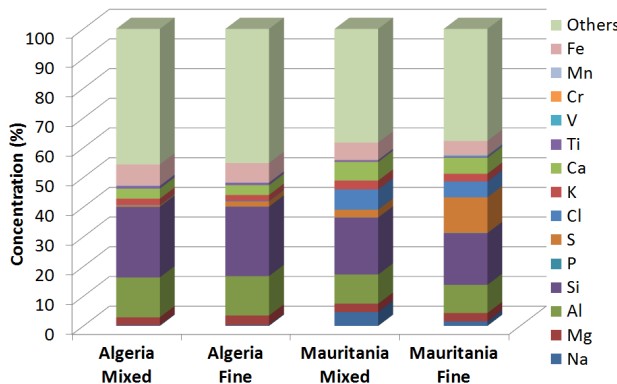

**Figure 12.** Mean mass concentration in percentage of the total mass of the aerosol particles obtained for each element for fine and mixed mode (fine+coarse) of the Saharan dust from Algeria and Mauritania by Energy Dispersive X-ray Fluorescence analysis (EDXRF).

Differences in the mean elemental composition can be seen between the supersites. Notably, SS2 in Mauritania has a higher concentration of sodium (Na) and chlorine (Cl), which suggest a "marine influence". Sulfur (S) is observed in larger concentration in the fine modes, also, the samples from the Mauritania site have a higher concentration of fine sulfate particles compared to the Algeria site. The Ca/Al ratio in Mauritania (0.57 and 0.64 for fine and mixed mode, respectively) is larger than in Algeria (0.25 and 0.25 for fine and mixed mode, respectively). That is in agreement with the Ca/Al ratio decreases





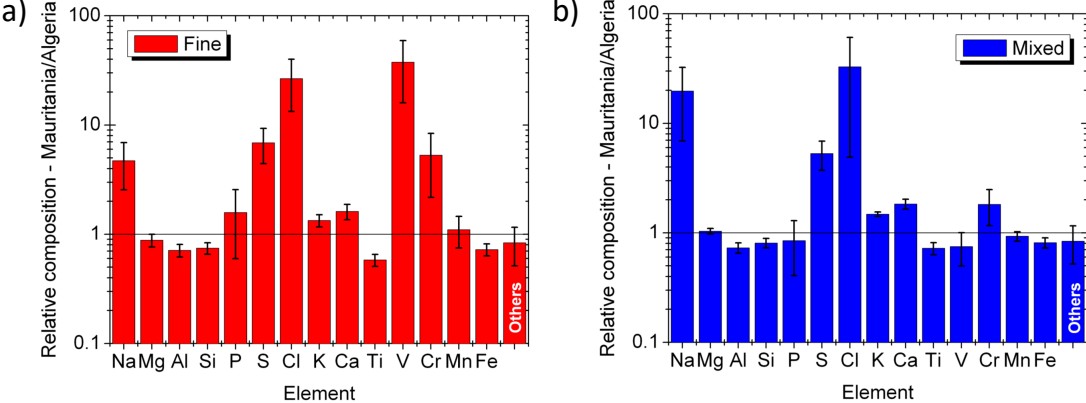

**Figure 13.** Relative elemental composition in logarithmic scale for each element between Algeria and Mauritania for the modes (a) fine and (b) mixed obtained by Energy Dispersive X-ray Fluorescence analysis (EDXRF).

observed in the Sahara from west to east described by Formenti et al. (2011). Source areas and composition has also been linked in Scheuvens et al. (2013), where they have found that (Ca+Mg)/Fe ratio is higher for sources areas coincident with SS2 in Mauritania and usually lower for sources areas coincident with SS1 in Algeria. The (Ca+Mg)/Fe ratio for Mauritania was found equal to (1.74 and 1.58 for fine and mixed mode respectively) and (1.0 and 0.87 for fine and mixed mode, respectively)

in Algeria. In addition to Na and Cl, the trace element vanadium (V) and Sulphur are observed in higher concentration in the fine mode at Mauritania. These differences are made clear by plotting the ratio of Mauritania's elemental composition relative to Algeria, as seen in Fig. 13.

# 6  Comparison between *in situ* and AERONET results

## 6.1  Comparison of scattering coefficient and total column aerosol optical thickness

Collocated measurements performed by an AERONET Cimel Sun photometer at SS1 in Algeria allowed us to compare our local ground-based measurements and derivations with those obtained from total column measurements. For example, the time series of the scattering coefficient presented in Section 3.2 was compared with AERONET (level 2.0) aerosol optical thickness (AOT) for the same period (Fig. 14). In part a), based on Marsham et al. (2013); Todd et al. (2013); Garcia-Carreras et al. (2015), we assume a 5 km deep Planetary Boundary Layer (PBL) with a constant vertical profile of dust in order to match

the units with the scattering coefficient measured by the nephelometer. We note that the clear conditions observed during the first days of the experiment are also apparent in the AERONET data. The AOT measurements are higher after 13 June, but AERONET total column measurements do not necessarily follow the fine details of the ground level observations, nor do the AOT measurements follow the full magnitude of large events. In addition, AERONET does not report level 2.0 data during some of the major dust storm events, likely due to its cloud screening process. Part b) shows that the correlation between ground



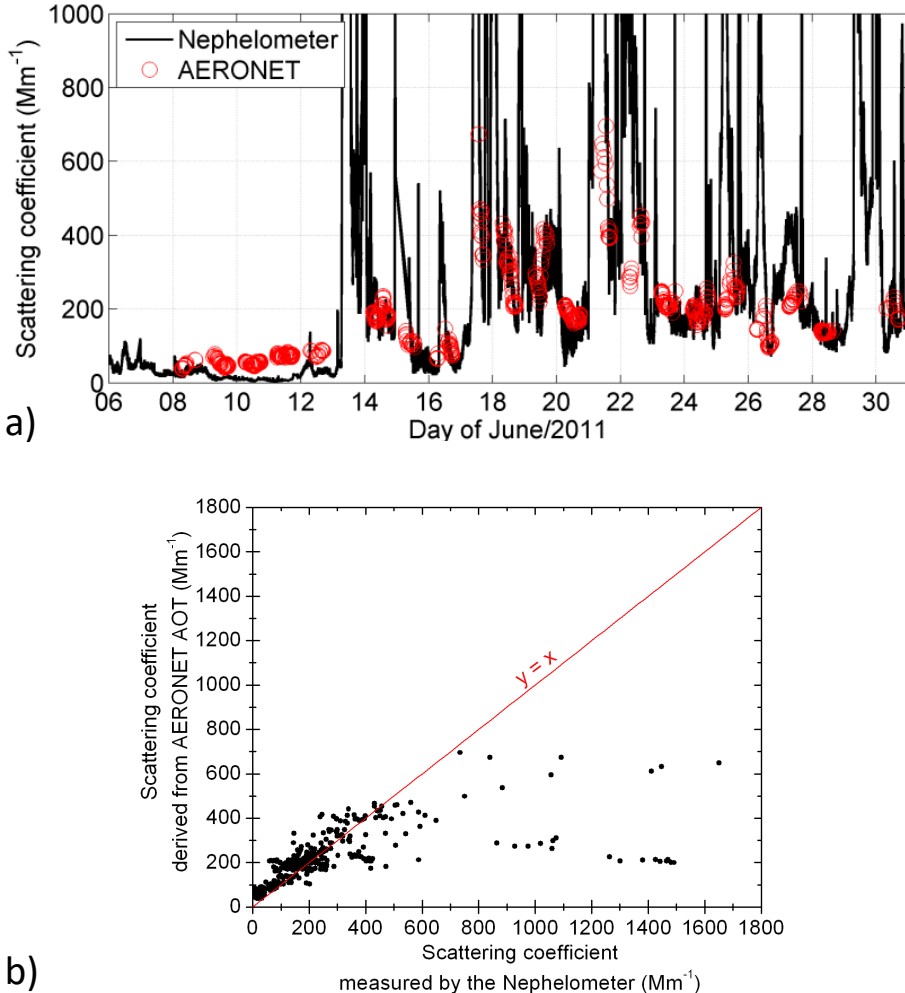

**Figure 14.** Intercomparison of AERONET total column measurements with ground–based measurement at SS1 in Algeria. a) Scattering coefficient measured by the nephelometer at the Fennec tower compared to AERONET AOT normalized by a factor of 5 km, b) Scatterplot of the scattering coefficient obtained by normalizing AERONET AOT by a factor of 5 km and measured by the nephelometer.

measurements and the total column is better when the scattering coefficient is below $\approx 400$ Mm$^{-1}$. During intense events of dust storms when the scattering coefficients reach higher values, ground based and total atmospheric column measurements do not maintain the same correlation, as the heavy dust loads occur during haboobs or low-level-jet breakdown and are not expected to occupy the full 5 km deep layer of the well mixed late afternoon PBL (Marsham et al., 2013). For example, it can
5   be seen in Ryder et al. (2013a) that during fresh dust events, as are likely to be dominant during the high scattering periods shown here, the vertical profile of dust is strongly dominated by loadings in the bottom 1-2 km of the atmosphere.





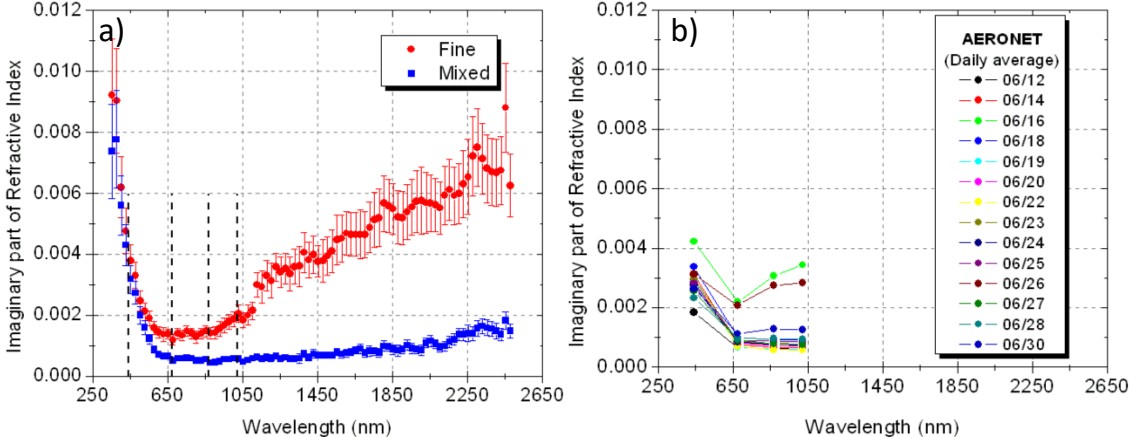

**Figure 15.** a) Spectral imaginary part of the refractive index for Saharan dust from SS1 in Algeria. The dashed lines indicate wavelengths of AERONET retrievals. b) AERONET (level 2.0) mean daily retrievals of imaginary part of refractive index from the collocated sun photometer. The outlier days exhibiting higher than average values correspond to 06/16 (bright green) and 06/26 (dark rust).

## 6.2  Comparison of imaginary part of the refractive index

The effective imaginary part of the refractive index derived from the mass absorption efficiency measurements, Fig. 11 a), were compared with AERONET retrievals for SS1 in Algeria retrieved during the Fennec campaign. Figure 15 a) shows our imaginary refractive index for fine and mixed mode where the dashed lines indicate the AERONET wavelengths and Fig. 15 b) shows the AERONET daily average of the imaginary refractive index for all days during the campaign, when available.

Notably, AERONET retrievals in Fig. 15 b) show a higher imaginary part of refractive index on days 16 and 26 of June, with a slightly increasing trend above 650 nm. AERONET retrievals of size distributions give higher concentrations of fine particles for these days, which is also observed in our *in situ* data in the time series of the fine mode mass fraction obtained by the ratio of the mass collected on the fine and the total (fine+mixed) modes, as shown in Fig. 16.

The very high concentrations of fine mode particles on these days indicate that fine particles dominated the AERONET retrieval of refractive index of the total column of aerosols. This agrees with the different spectral signatures we observe in our derivations of the refractive index in the fine and mixed modes at SS1–Algeria. The comparison of our refractive index with AERONET retrievals at SS2 in Mauritania was not possible, because AERONET does not have measurements nearby. The closest AERONET to our SS2 station was at Zourete, 290 km away. It is interesting to note that our retrievals of refractive index for the fine and the mixed modes from SS2 in Mauritania follow the same spectral dependence as the fine mode in Algeria, as seen in Fig. 11 b). In addition to that, it is important to note that the fraction of fine particles in the mixed mode in SS2 is much larger than in Algeria, as seen in the particle number distributions in the top panels of Figs. 9 a) and (b) and also in the mean mass concentration shown in Figs. 12 a) and (b). This dominance of fine particles in the mixed mode may explain why we found the same spectral dependence of the refractive index in both fine and mixed modes in Mauritania.





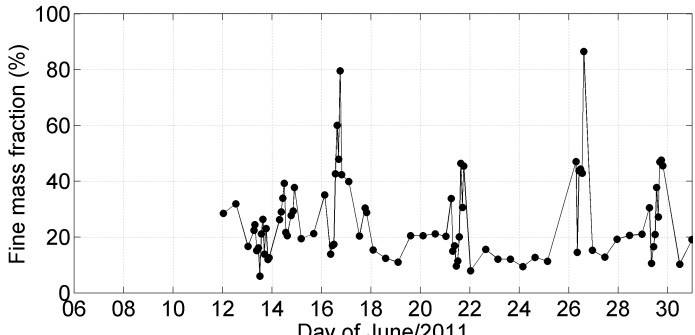

**Figure 16.** Fine mass fractions were obtained by dividing the mass concentrations of the fine mode to that of the total (fine+mixed) modes collected on filters using the LACO aerosol sampling station during the Fennec experiment at SS1 – Algeria. Two main peaks were observed on days 16 and 26 of June indicating the lower concentration of coarse particles.

## 7   Discussion and Conclusions

Real time *in situ* measurements and *in situ* filter collection were obtained from two Fennec supersites in the central Sahara, one in Algeria and the other in Mauritania. In Mauritania, analysis of the samples collected with the LACO Aerosol Sampling Station shows the presence of low density particles with aerodynamic diameters larger than 10 μm in some of days. These

particles are not typical of the dust observed in most of the filters. They have more complex shapes, lower density and can be easily deteriorated during SEM analysis. On the other hand, the low density particles were not observed in Algeria. Even when confining the analysis of Mauritania size distribution to only samples of Group 1 (without the large low density particles), we find higher concentrations of fine particles and low number of coarse particles in the mixed mode. In contrast, the mixed mode in Algeria presents a more pronounced number of coarse particles. Differences between the two sites were also seen in

the elemental composition obtained by XRF analysis. The most notable differences were the higher concentrations of Na, Cl, and S in the samples collected in Mauritania. The ratio of some key chemical components, such as Ca/Al and (Ca+Mg)/Al follows expected variation associated with sources areas where the sites are located. This variation of dust chemical and optical properties at these two sites suggest that not all aerosol found over the Sahara or transported from the Sahara can be modeled as "typical" dust. This is an important result because it corroborates previous work that Saharan aerosol exhibits different optical

and microphysical properties. In this work we see this variation even in the central Sahara, where there have been no previous measurements of this type over the past decade. More studies are needed to fully characterize the Saharan regional variability, as this information should be captured by dust aerosol models attempting to simulate Saharan aerosol and by remote sensing algorithms measuring dust properties from space.

The spectral imaginary refractive index derived for the fine mode in both sites shows a similar and distinctive bow-shaped

spectral dependence. Not only does the imaginary part of the refractive index increase sharply at the shortwave end of the spectrum, as expected, but the value also increases from 650 nm towards the shortwave infrared. Wagner et al. (2012) also derived spectral imaginary part of the refractive index. Their spectral range spanned 300–950 nm, and despite the relatively





curtailed spectral range, there is no apparent bow-shape in their results. However, this bow-shape signature is seen in other previous work by Balkanski et al. (2007) and references therein. It is also seen in the AERONET retrievals at SS1 for days when the fine mode aerosol dominates. The bow shape was also seen in spectral remote sensing retrievals of aerosol absorption over the Sahara at some locations (Wells et al., 2012). The link between the bow shape and fine mode particles may also explain

why this spectral signature is not found in the mixed mode at the Algeria station, which has a higher concentration of coarse particles in its mixed mode. One of the major conclusions of this work is the identification of the bow-shaped spectral signature in the imaginary part of the refractive index of fine mode particles over the Sahara.

The values of the imaginary part of the refractive index for fine mode dominated aerosols at both stations are 0.0030i to 0.0045i, 0.0015i to 0.0030i, 0.0015i to 0.0025i, and 0.0018i to 0.0030i, for wavelengths of 450 nm, 550 nm, 650 nm and

850 nm, respectively. Uncertainties of the imaginary part of the refractive index for the fine mode were estimated to be a maximum of 25%. For the mixed mode in Algeria where coarse mode particles dominate, the imaginary refractive index of the mixed mode is nearly constant above 650nm. Here, the imaginary refractive index is 0.0030i, 0.0005i, 0.0005i, 0.0005i for wavelengths of 450 nm, 550 nm, 650 nm and 850 nm, respectively, with maximum uncertainties on the order of 25%. These results are less than half of the values retrieved by Wagner et al. (2012) for their SAMUM samples. However, the imaginary

part of the refractive index of the fine mode is consistent with values inferred from remote sensing observations (Colarco et al., 2002, 2014; Kim et al., 2011).

The *in situ* measurements of scattering and absorption coefficients in Algeria allowed us to calculate the temporal single scattering albedo of the Saharan dust at 670 nm. Our values of single scattering albedo varied from 0.96 to 1.0 and are 0.01 to 0.02 systematically higher than the values measured by AERONET, although still compatible within uncertainties from both

methods.

Ryder et al. (2013a) present results of dust optical properties measured and derived during Fennec from aircraft over northern Mauritania and North West of Mali. Differences between these airborne measurements and our ground-based results appear striking at first glance. For example, Ryder et al. (2013a) show that directly measured SSAs at 550 nm representing the accumulation mode (d< 2.5 μm) ranged from 0.91 to 0.99 with a mean of 0.97. Once the full size distribution including the

coarse mode was included, Mie scattering calculations showed that the SSA at 550 nm dropped to 0.86–0.97 (mean 0.92) when a refractive index of 1.53-0.001i was assumed. In contrast, our ground-based measurements for SSA at 670 nm for d< 10 μm are 0.99 to 1.0. From typical spectral signatures of dust absorption (e.g. Figs. 11 and 15) we would expect much less absorption and higher SSA values in our measurements than from Ryder et al. (2013a) based on the differences in wavelength alone. Adjusting the Fennec airborne measurements to 670 nm might increase the SSA values by around 0.1 (e.g. Otto et al.

(2009)). This would push the Fennec airborne accumulation-only mode SSA values into the region of those presented here but this is not the case for the SSA values representing the full size distribution.

There are other factors contributing to the differences between the airborne and ground-based results. The size distribution measured by these aircraft observations showed a strong coarse mode, with effective diameter covering 2.3–19.4 μm and coarse mode volume median diameter 5.8–45.3 μm. These are much larger particles than those collected by the ground-based

instruments for analysis, not because larger particles did not exist at ground level, but because the ground instruments purposely





removed particles larger than 10 μm diameter with an aerodynamic impactor. The absence of the large particles in our analysis can explain some of the divergence between the ground-based and airborne SSA results that the wavelength differences cannot. These differences in instrumentation characteristics (wavelength and size cut-off) make conclusions about real differences in optical properties between near-ground and elevated dust difficult. However, we note that AERONET retrieved total column

ambient SSA values at 675 nm at SS1 are most frequently in the 0.975–0.99 range, which overlaps the ground-based and airborne values for smaller size ranges, and suggests that the contribution of the largest particles to total column values is small.

Overall, the results show that the dust of the central Sahara measured during Fennec at ground-level shows low absorption characteristics and exhibits a distinctive spectral bow-like shape unlike other more absorbing measurements, especially other

measurements of pure dust samples from the SAMUM experiment in the northwest edge of the desert. The bow-like shape, with increased absorption in the shortwave infrared may hold consequences for calculations of spectrally integrated aerosol radiative effects. We also find size dependence in the dust absorption spectral signature that has not been noted previously and may correspond to other size-dependent characteristics such as aspect ratio (Kandler et al., 2009). Like other studies we find distinctive differences in the composition and optical characteristics of the dust from the two Fennec sites, pointing once again

to the fact that not all Saharan dust is the same, even pure dust isolated from biomass burning. Thus, measurement campaigns like Fennec strategically placed in various desert locations continue to be necessary in order to narrow the uncertainties in characterizing dust microphysical and optical properties, which will place constraints on attempts to model the transport, radiative and climate effects of this important aerosol type.

*Acknowledgements.* We thank the entire Fennec team, specially the authors would like to thank M. Salah-Ferroudj, B. Ouchene, A. Ouladichir,

and A. Saci from The Met office in Algeria and Sidate Deyane in Mauritania for their hard working in the installation and operating of the Fennec instruments during the Fennec campaign. We are thankful to the engineering team at LACO-UMBC for their support in all areas of preparation and assembling the Aerosol Sampling Stations and the Integrating Nephelometer sent to Sahara for the Fennec campaign, especially, Dominik Cieslak, John Hall, Kevin Townsend, Tim Kuester. We thank the scientific and technical support of Manfredo H. Tabacniks, Alexandre Lima Correa and Ana Lucia Loureiro from the University of São Paulo. Adriana Rocha-Lima thanks all the members of her PhD

committee for the incentive and ideas in how to improve this work: Raymond Hoff, Zhibo Zhang, Nickolay Krotkov, Andreas Beyersdorf, and Laszlo Takaks. We thank Martin Todd for his effort in establishing and maintaining the Bordj Badji Mokhtar AERONET site during the Fennec campaign. J. Marsham was funded by the NERC Fennec (NE/G017166/1) and SWAMMA (NE/L005352/1) projects. R. Washington was funded by NERC Fennec (NE/G016283/1)project.



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
