# Peer review of "A detailed characterization of the Saharan dust collected during the Fennec Campaign in 2011: *in situ* ground-based and laboratory measurements"

_Atmospheric Chemistry and Physics, 2017_

## Referee Comment (RC1) · Anonymous Referee #1 · 11 Jul 2017

Review of "A detailed characterization of the Saharan dust collected during the Fennec Campaign in 2011: in situ ground-based and laboratory measurements" by Rocha-Lima et al.

The publication describes spectral optical scattering/absorption and mass concentration measurements performed at two locations in Mauritania and Algeria during an intensive operation period of the Fennec campaign in 2011. Moreover, it includes supplementary measurements of size distribution, particle density and bulk chemical composition. For the optical measurements the authors use a new combined approach of

nephelometry and filter-based reflectance measurements, which they describe shortly. They present time series of dust concentrations and optical properties like single scattering albedo and imaginary part of refractive index, which are set into context with third party and literature data. They conclude that fine mode dust can be at times dominate the dust optical properties and that Saharan dust is not uniform and should not be considered as homogeneous, e.g. for later modeling purposed.

The paper is well-written and clearly structured, references are made where appropriate. The new technique is not described in detail, some more information or a reference would be helpful here.

Apart from that, only some minor remarks are to be regarded from my point of view:

P6L4 and P15L8-10: Aerodynamic diameter is defined for spheres, and non-spherical particle shapes like for dust will lead to larger aerodynamic diameters, so the 6.1 $\mu$m cut-off is most probably a minimum estimate. Reversely, it might not only the density of the particles leading to a different aerodynamic diameter.

Figs. 3 and 4: I suggest combining into one figure with single time axis to facilitate comparison.

P11: I suggest moving the method description to a single chapter before results (as in the pages before) are presented. Also, more information on the new technique should be provided (e.g., a calibration plot of a material with known optical properties).

P15L3-17: Damage by the electron beam and low aerodynamic diameter indicates biological debris. Is there no EDX spectrum available?

P15L28: Numbers of stages / pore size exchanged.

Caption figure 9 and other places: SEM sizes refer usually to projected area equivalent diameter. Although geometric diameter is not totally wrong, I suggest using the more precise term.

P18L15-16: In which cases dust particles are commonly spherical?

Fig. 12: Comparison with total aerosol mass would make more sense, if oxide weights would be used, where applicable.

P21L5-7: How significant is the vanadium (and chromium signal), as it might be tracer for certain industrial activities?

---

## Referee Comment (RC2) · R. L. Miller (Referee) · 27 Aug 2017

This article characterizes Saharan dust collected during the FENNEC campaign in June 2011. Measurements at two sites (Algeria, Mauritania) are used to derive various optical properties of local aerosol samples. Filter samples were subsequently analyzed in a laboratory to provide additional information about particle size and elemental composition. The article contributes to the literature characterizing the physical and chemical properties of aerosols (mainly dust) within the Sahara. These studies are especially important for recent modeling work that attempts to characterize regional variations in

the mineral content of dust particles. My comments are mainly requests for clarification. I am recommending acceptance subject to minor revision. If the authors have any questions about my review, they can contact me at ron.l.miller@nasa.gov.

1) The 'mixed size' sample of particles collected by the filter captures a range of particle diameters, including smaller particles that overlap in size with that of the 'fine' sample (whose diameters are less than 5 um). This means that the mixed-size sample will differ from the actual size distribution of the ambient aerosols. In particular, the mixed-sized sample should have fewer fine particles than are in the air, because some of these passed through the filters into the fine sample. This means that aerosol properties that depend upon the distribution of particle diameter (like SSA), will differ from the actual ambient values that might be measured by AERONET (e.g.). This makes it difficult for modelers (or other measurement scientists whose size distribution will differ) to compare their SSA to the values reported in this study. I suggest that in the abstract and conclusions, the authors give emphasis to properties like index of refraction that are less dependent than SSA upon particle size, since the index can be more directly compared to values from other studies.

2) The description of the derivation of some of the optical properties (Section 4.3 and 5) is intricate. I occasionally had difficulty keeping track of what measurements were used to derive a particular optical property. The addition of a table relating a theoretical property (like SSA or absorption coefficient or index of refraction) to the specific measurements used, along with the temporal resolution of the property (that depends upon the measurement with the lowest temporal resolution) would be appreciated.

Minor Comments:

p.1 line 11, 13: The authors should give a specific diameter range for 'fine' amd 'mixed size'

p.2 line 7: 'acts to cool the planet'. The direct radiative forcing at TOA by dust is uncertain (as noted by the authors). Kok et al. note that the forcing could actually be

positive (leading to warming of the planet). This possibility should be acknowledged.

Kok, J.F., D.A. Ridley, Q. Zhou, R.L. Miller, C. Zhao, C.L. Heald, D.S. Ward, S. Albani, and K. Haustein, 2017: Smaller desert dust cooling effect estimated from analysis of dust size and abundance. Nature Geosci., 10, no. 4, 274-278, doi:10.1038/ngeo2912.

line 8, 10: replace 'radiative forcing' with 'direct radiative forcing'?

p.4 line 15: define 'microphysical distribution' more specifically?

p.5 line 4: for the benefit of non-specialist readers, define the diameter range of the accumulation mode?

line 14: 'a technique applied' please provide some description of this technique

Table 1: could you clarify 'Sampling period'? Is each measurement a time-average over this period? Is this period the temporal resolution of each measurement?

p.8 line 20: 'These peaks are asociated with the sudden moistening convective events...' Allen et al (2013) attribute at least part of the event of June 18 to the breakdown of a low-level jet (see their Table 1 and their Figure 1c). The same comment applies to p.10 line 9: 'follows a moistening event'.

Allen, C. J. T., R. Washington, and S. Engelstaedter (2013), Dust emission and transport mechanisms in the central Sahara: Fennec ground-based observations from Bordj Badji Mokhtar, June 2011, J. Geophys. Res. Atmos., 118, 6212–6232, doi:10.1002/jgrd.50534

p.10 line 4: 'On June 22, the detection scale of the nephelometer was reconfigured...' Is this why Figure 4 looks slightly different from the nephelometer times series in Figure 1a of Allen et al. (2013). (It may be unreasonable to expect the authors to know the answer here, but if they do, it would be useful for readers familiar with the FENNEC literature.)

p.11 line 8: 'a large range of geometries' What does 'geometries' refer to? The interior

of the nephelometer?

line 12: 'the reflectance of the Nuclepore filter cannot be increased' Could you put this in plainer language? Do you mean that the filters only get dirtier and less reflective as the air flows through? (For a few events in Figure 5b, the reflectivity at 670 nm initially increases. Is this a measure of instrument error?)

line 19: what is delta t in this case? 45 seconds, corresponding to the sampling period of the reflectometer? In general, this page is very technical and difficult to read, although instrument scientists will have an easier time.

line 25: insert 'temporal' before 'subset'?

p.12 line 11: 'These results show variation along this period from 0.96 to close to 1, with a mean value around 0.995...' The low values near 0.96 in Figure 5d are short-lived. Are the values just noise or is there a physical cause?

line 12: 'systematically higher than the values for Saharan dust found in the literature at this wavelength' Please give examples of such studies.

Figure 5: Label each panel with a, b, c or d to correspond to the caption.

Figure 6: what do the different colored lines correspond to? Samples from different days?

p.14 line 11 'This strong spectral dependence is what causes the dust to appear brown to our eye.' This is an interesting comment. Could you elaborate by telling me what primary colors brown corresponds to? If I am interpreting the figure correctly, dust absorbs blue and some of green, allowing red to be scattered back to our eyes. (I will always regret not taking an optics class as an undergraduate :)

line 14: 'three groups of samples were identified' Were the Mauritania groups divided based on qualitative inspection or was an objective criterion used?

p.15 line 32: 'Other studies also show a decrease in coarse mode fraction as sampling

moves towards aged dust and away from fresh dust near sources.' Please be specific and state which of the measurement sites is considered to be closer to the upwind sources.

p.16 line 3: 'A comparison between this geometrical distribution...' This sentence is unclear. How was the impactor efficiency taken into account in this study? Could the authors explain briefly why this matters? (This is probably obvious to an instrument scientist.)

p.18 line 5: 'Once the instruments arrived back at UMBC, dust deposited on the instrument surfaces was gently collected using a brush and sieved using a 45 $\mu$m mesh grid.' Does mechanical brushing break up larger aggregate particles, modifying the aerosol size distribution that was latter characterized using the SEM?

p.19 line 3: 'Figure 11 a) shows that the imaginary part of the complex refractive index of Saharan dust from Algeria has significant spectral differences between fine and mixed mode.' The authors should explain how the refractive index was calculated for the mixed mode, if the T-matrix code doesn't converge (p.18, line 20). (The answer appears to be in the caption of Figure 11, but this should be described in the text.) For samples from Mauritania, the use of Mie theory seems to introduce an uncertainty comparable to the measurement uncertainty, given the sensitivity of the fine sample index to this assumption. This should be discussed.

line 5: 'For longer wavelength the values diverge considerably...' Why is this? For pure materials, the index of refraction should be independent of particle size. Is this divergence evidence that the fine and mixed samples are comprised of different minerals?

Figure 12: I find it difficult to derive much quantitative information from this figure. Could the authors replot these four cases as four bar graphs, as in Figure 13?

Section 5.6: ('EDXRF analysis of Saharan dust') Are there any differences in elemental composition with respect to particle size? This is an important question for modelers

trying to reproduce the hematite composition in soil maps. (e.g. Scanza et al ACP 2015, Perlwitz et al ACP 2015). For example, Perez et al (2016) noted that Fe (from hematite) is mainly independent of particle size at Izana (just downwind of these sites), contrary to some soil mineral atlases that restrict it to larger particle sizes. A figure similar to Figure 13 showing the difference of elemental composition between the fine and mixed modes would be helpful to address this question.

Perez Garcia-Pando, C., R. L. Miller, J. P. Perlwitz, S. RodríÅguez, and J. M. Prospero (2016), Predicting the mineral composition of dust aerosols: Insights from elemental composition measured at the IzanÌČa Observatory, Geophys. Res. Lett., 43, doi:10.1002/ 2016GL069873.

Perlwitz, J. P., Perez Garcia-Pando, C., and Miller, R. L.: Predict- ing the mineral composition of dust aerosols – Part 1: Representing key processes, Atmos. Chem. Phys., 15, 11593–11627, doi:10.5194/acp-15-11593-2015, 2015.

Scanza, R. A., Mahowald, N., Ghan, S., Zender, C. S., Kok, J. F., Liu, X., Zhang, Y., and Albani, S.: Modeling dust as component minerals in the Community Atmosphere Model: development of framework and impact on radiative forcing, Atmos. Chem. Phys., 15, 537–561, doi:10.5194/acp-15-537-2015, 2015.

p.21 line 15: '...or do the AOT measurements follow the full magnitude of large events.' This could be because cold pools often arrive at night (Allen et al 2013), when the sun photometer is not measuring.

p.22 line 5: 'For example, it can be seen in Ryder et al. (2013a) that during fresh dust events...' This is interesting. Here, the authors seem to be arguing that discrepancies between AERONET and the nephelometer are due to departures of the height of the dust layer from the assumed value of 5 km. Can the authors estimate the frequency of this underestimate by AERONET compared to the frequency of missing retrievals because the dust concentration is so high that the aerosol layer is mistaken for a cloud?

Interactive
comment

p.24 line 12: "...follows expected variation associated with sources areas where the sites are located.' Do the authors mean specifically that the Mauritania site is more influenced by marine aerosols?

p.25 line 21: 'Ryder et al. (2013a) present results of dust optical properties measured and derived during Fennec from aircraft over northern Mauritania and North West of Mali. Differences between these airborne measurements and our ground-based results appear striking at first glance.' SSA depends upon the size distribution (as the authors note below). How much does this contribute to the different compared to possible differences in composition?

p.25 line 6: 'However, this bow-shape signature is seen in other previous work by Balkanski et al. (2007) and references therein.' I believe that a bow shape is also implicitly present in some models (e.g. GISS: c.f. Tegen and Lacis 1996; Miller et al JGR 2006) that interpolate the index of refraction between measurements in the visible (Patterson et al 1977 or Sinyuk et al 2003) and IR (Volz 1973). The imaginary part of the index is small within the visible (Sinyuk et al 2003), but rises as a result of interpolation to meet higher values in the IR around 3 um (Volz 1977).

Miller, R. L., et al. (2006), Mineral dust aerosols in the NASA Goddard Institute for Space Sciences ModelE atmospheric general circulation model, J. Geophys. Res., 111, D06208, doi:10.1029/2005JD005796.

Patterson, E. M., D. A. Gillette, and B. H. Stockton (1977), Complex index of refraction between 300 and 700 nm for Saharan aerosols, J. Geophys. Res., 82, 3153–3160.

Sinyuk, A., O. Torres, and O. Dubovik, Combined use of satellite and surface observations to infer the imaginary part of refractive index of Saharan dust, Geophys. Res. Lett., 30(2), 1081, doi:10.1029/ 2002GL016189, 2003.

Tegen, I., and A. A. Lacis (1996), Modeling of particle influence on the radiative properties of mineral dust aerosol, J. Geophys. Res., 101, 19,237 – 19,244.

Volz, F. E. (1973), Infrared optical constants of ammonium sulfate, Sahara dust, volcanic pumice and flyash, Appl. Opt., 12, 564–568.

p.26 line 13: 'may correspond to other size-dependent characteristics such as aspect ratio' Or mineral composition? For example, MoosmuÌĹller et al (2012) show that for aerodynamic diameters less than 2.5 um, SSA from African dust particles is linearly related to the elemental fraction of iron (which they attribute to hematite).

Moosmuller, H., J. P. Engelbrecht, M. Skiba, G. Frey, R. K. Chakrabarty, and W. P. Arnott (2012), Single scattering albedo of fine mineral dust aerosols controlled by iron concentration, J. Geophys. Res., 117, D11210, doi:10.1029/2011JD016909.

---

## Author Comment (AC1) · 24 Oct 2017

We would like to thank both referees for their constructive comments. Please find below the point-by-point response for each of the questions raised by the referees. Additionally, you can see a revised version of the manuscript attached where all modifications were highlighted in bold for easier identification.

**Response to the Anonymous Referee #1**

**Review of "A detailed characterization of the Saharan dust collected during the Fennec Campaign in 2011: in situ ground-based and laboratory measurements" by Rocha-Lima et al. 2017.**
**The publication describes spectral optical scattering/absorption and mass concentration measurements performed at two locations in Mauritania and Algeria during an intensive operation period of the Fennec campaign in 2011. Moreover, it includes supplementary measurements of size distribution, particle density and bulk chemical composition. For the optical measurements, the authors use a new combined approach of nephelometry and filter-based reflectance measurements, which they describe shortly. They present time series of dust concentrations and optical properties like single scattering albedo and imaginary part of refractive index, which are set into context with third party and literature data. They conclude that fine mode dust can be at times dominate the dust optical properties and that Saharan dust is not uniform and should not be considered as homogeneous, e.g. for later modeling purposed.**
**The paper is well-written and clearly structured, references are made where appropriate. The new technique is not described in detail, some more information or a reference would be helpful here.**

Thank you for your feedback. The laboratory techniques used in this study were previously described in more details in [Martins et al., 2009; Rocha-Lima et. al. 2014]. For that reason, we presented only a simplified description of these methods giving the reference for those papers when appropriated. We added this information in (P3L22) of the Introduction:

*"In situ measurements of Saharan dust were complemented with laboratory analyses for the characterization of their optical properties using the methods presented in Martins et al. (2009) and Rocha-Lima et al. (2014)."*

The description of the instruments and methods related to in-situ and laboratory measurements are presented in Sections 3, 4, and 5. We included additional information about the techniques along these sections and we also emphasized that more details can be found in these references.

Martins, J. V., Artaxo, P., Kaufman, Y. J., Castanho, A. D., and Remer, L.: Spectral absorption properties of aerosol particles from 350-2500nm, Geophys. Res. Lett. 36, L13810, doi:10.1029/2009GL037435, 2009.

Rocha-Lima, A., Martins, J. V., Remer, L. A., Krotkov, N. A., Tabacniks, M. H., Ben-Ami, Y., and Artaxo, P.: Optical, microphysical and compositional properties of the Eyjafjallajokull volcanic ash, Atmos. Chem. Phys., 14, 10649-10661, doi:10.5194/acp-14-10649-2014, 2014.

**Apart from that, only some minor remarks are to be regarded from my point of view: P6L4 and P15L8-10: Aerodynamic diameter is defined for spheres, and non-spherical particle shapes like for dust will lead to larger aerodynamic diameters, so the 6.1 µm cut-off is most probably a minimum estimate. Reversely, it might not only the density of the particles leading to a different aerodynamic diameter.**

We agree with the point raised by the referee. The definition of aerodynamic size is for a spherical particle with density equal to 1. As the referee noticed, both the density and the particle's shape play a role in the conversion from aerodynamic sizes to geometrical sizes. We modified this part of the text for clarity (P6L3):

*"Without taking the shape of the particles into account, for particles of density around 2.6g/cm³, such as dust, this is approximately equivalent a cut size of 50% to spherical particles with a dimeter of 6.1µm."*

**Figs. 3 and 4: I suggest combining into one figure with single time axis to facilitate comparison.**
We combined the figures 3(a) and 4 as suggested. The scattering coefficient is now overlapped with the time series of the mass concentration in Figure 3(a) for Algeria:

[Figure]

The comparison shows that mass concentration and scattering coefficient are correlated most of the time. However, some differences are observed, most importantly being that short duration events of high concentrations are not captured by the Aerosol Sampling Station. This happens due to differences in the time resolution of these measurements. The mass concentration is averaged over periods of 6 hours, while the nephelometer had a time resolution of 45 seconds. This information was added in Section 4.1 (P9L27):

*"It is important to note that dust events of short duration observed in the scattering coefficient are not captured by the measurements of mass concentration. This happens due to different time resolution of the instruments used for these two measurements. The mass concentration is averaged over 6 hours, while the nephelometer had a time resolution of 45 seconds."*

**P11: I suggest moving the method description to a single chapter before results (as in the pages before) are presented. Also, more information on the new technique should be provided (e.g., a calibration plot of a material with known optical properties).**

We considered this suggestion but we respectfully would like to keep the same structure. We prefer to keep the methodology from *in situ* and laboratory measurements in different sections. The techniques and methods from *in situ* measurements performed during the field campaign are presented in "*Section 3. Instrument and Sites*" and "*Section 4. Times Series Dust Characterization*". Following that, we present the techniques and methods used for the derivation of other microphysical and spectral optical properties from laboratory measurements in "*Section 5. Spectral Imaginary Refractive of Dust Index*".

The description and validation of the method used for derivation of the mass absorption efficiency was previously discussed in Martins et al. (2009). In this work, the mass absorption efficiency of Monarch71 particles (manufactured by Cabot Corporation) derived using this technique shows good agreement with other measurements and with the theoretical modeling. A complete description of the techniques used for the derivation of the imaginary refractive index from spectral optical measurements is presented in Rocha-Lima et al. (2014). We included these references when appropriated along the manuscript. We also included in Section 4.2 a sentence explicitly saying that more details about these techniques can be found in the references above (P11L15):

*"A detailed description of this method can be found in Martins et al. (2009) and Rocha-Lima et al. (2014)."*

Also note that the spectral imaginary refractive index derived in Rocha-Lima et al. (2014) was compared with literature data obtained by different methods. For instance, Carn and Krotkov (2016) make a compilation of global data for the imaginary part of the refractive index of volcanic ashes. That work shows that the results obtained with this technique are in good agreement with literature data in the UV range.

More recently Vogel et al. (2017) compared the measurements of the spectral imaginary refractive index presented in Rocha-Lima et al. (2014) with their UV-NIR measurements showing high correlation in the visible and NIR wavelengths range and less correlation in the UV range.

Carn, S. A. and Krotkov, N. A., Chapter 12 - Ultraviolet Satellite Measurements of Volcanic Ash, In Volcanic Ash, edited by Shona Mackie, Katharine Cashman, Hugo Ricketts, Alison Rust and Matt Watson, Elsevier, 217--231, ISBN 9780081004050, doi:10.1016/B978-0-08-100405-0.00018-5, 2016.

Vogel, A., S. Diplas, A. J. Durant, A. S. Azar, M. F. Sunding, W. I. Rose, A. Sytchkova, C. Bonadonna, K. Krüger, and A. Stohl: Reference data set of volcanic ash physicochemical and optical properties, J. Geophys. Res. Atmos., 122, 9485-9514, doi:10.1002/2016JD026328, 2017.

**P15L3-17: Damage by the electron beam and low aerodynamic diameter indicates biological debris. Is there no EDX spectrum available?**

There might be indeed the possibility of biological debris in the samples. Unfortunately, we did not further investigate possible biological materials in our samples.

**P15L28: Numbers of stages / pore size exchanged.**

We corrected this part of the text (P17L2):

*"Analysis included both the 1$^{st}$ stage filters with pore size 5 μm and the 2$^{nd}$ stage filters with pore size 0.4 μm as shown in Section 3.2."*

**Caption figure 9 and other places: SEM sizes refer usually to projected area equivalent diameter. Although geometric diameter is not totally wrong, I suggest using the more precise term.**

We replaced this term as suggested (see Figure 8).

**P18L15-16: In which cases dust particles are commonly spherical?**

As shown in Figure 9, the aspect ratio of the analyzed particles is generally greater than one. Therefore, the dust particles sampled in this work are typically non-spherical. The message we wanted to give in this sentence is that in addition to using Mie code in our work, we also used the T-matrix code to evaluate the effects of the shape of these particles in our retrieval of the imaginary part of the refractive index. We modified this sentence in the manuscript for clarification (P18L10):

*"However, the aspect ratio distribution of the particles shows that they are typically non-spherical. Therefore, in addition to the Mie code, the extended-precision T-matrix code (Mishchenko et al., 1996) assuming randomly oriented ellipsoidal particles was used with a modified gamma distribution fitted to the measurements."*

**Fig. 12: Comparison with total aerosol mass would make more sense, if oxide weights would be used, where applicable.**

We considered this possibility; however, this comparison would require assumptions for the oxidation state of the elements. In addition, low-Z elements other than oxygen are also not seen by EDXRF. Therefore, we prefer to compare directly the measured elemental composition of the samples, without any assumption of oxidation states of the elements or concentration of low-Z elements in the samples.

**P21L5-7: How significant is the vanadium (and chromium signal), as it might be tracer for certain industrial activities?**

Higher concentrations of vanadium in the fine mode in Mauritania indicate a source of pollution nearby this supersite. This information was added to the text (P21L6):

*"[...] trace elements usually related to pollutants as vanadium (V), chromium (Cr), phosphorus (P), and sulfur (S) are observed in higher concentration in the fine mode at Mauritania. The concentrations of vanadium in Algeria and in the mixed mode in Mauritania were relatively low, on the order of 100 ppm. These levels of vanadium are compatible with the natural abundance of this element in Earth's crust [Byerrum, 1991; ATSDR, 2012]. However, the concentration in the fine mode at Mauritania was on average 3000 ppm, indicating a significant contribution of anthropogenic sources nearby this supersite."*

ATSDR (Agency for Toxic Substances and Disease Registry): Toxicological profile for Vanadium. Atlanta, U.S. Department of Health and Human Services, Public Health Service, 2012.

Byerrum, R.U.: Vanadium. In: Merian E, ed., Metals and their compounds in the environment. Weinheim, Germany: VCH, 1289-1297, 1991.

**Response to the Referee #2: Ron Miller**

**This article characterizes Saharan dust collected during the FENNEC campaign in June 2011. Measurements at two sites (Algeria, Mauritania) are used to derive various optical properties of local aerosol samples. Filter samples were subsequently analyzed in a laboratory to provide additional information about particle size and elemental composition. The article contributes to the literature characterizing the physical and chemical properties of aerosols (mainly dust) within the Sahara. These studies are especially important for recent modeling work that attempts to characterize regional variations in the mineral content of dust particles. My comments are mainly requests for clarification. I am recommending acceptance subject to minor revision. If the authors have any questions about my review, they can contact me at ron.l.miller@nasa.gov.**

Thank you for your feedback and the positive evaluation.

**1) The 'mixed size' sample of particles collected by the filter captures a range of particle diameters, including smaller particles that overlap in size with that of the 'fine' sample (whose diameters are less than 5 um). This means that the mixed-size sample will differ from the actual size distribution of the ambient aerosols. In particular, the mixed-sized sample should have fewer fine particles than are in the air, because some of these passed through the filters into the fine sample. This means that aerosol properties that depend upon the distribution of particle diameter (like SSA), will differ from the actual ambient values that might be measured by AERONET (e.g.). This makes it difficult for modelers (or other measurement scientists whose size distribution will differ) to compare their SSA to the values reported in this study. I suggest that in the abstract and conclusions, the authors give emphasis to properties like index of refraction that are less dependent than SSA upon particle size, since the index can be more directly compared to values from other studies.**

We agree that the SSA is a size-dependence quantity and this is important when comparing SSA derived using different techniques. We have added a sentence in Sec. 4.2 to explicitly acknowledge that (P12L21):

*"It is important to note that SSA is a size-dependent quantity, and comparisons with other measurements should take into account size differences, as discussed in Sec. 7."*

We still would like to mention those quantities in the abstract and conclusion as they are part of our results.

**2) The description of the derivation of some of the optical properties (Section 4.3 and 5) is intricate. I occasionally had difficulty keeping track of what measurements were used to derive a particular optical property. The addition of a table relating a theoretical property (like SSA or absorption coefficient or index of refraction) to the specific measurements used, along with the temporal resolution of the property (that depends upon the measurement with the lowest temporal resolution) would be appreciated.**

We added a new table (Table 2) in Section 3.2 summarizing the measurements that were used to derive a particular optical property.

**Minor Comments:**
**p.1 line 11, 13: The authors should give a specific diameter range for 'fine' and 'mixed size'**

The fine mode contains particles with a maximum size of 5μm, and a volume distribution peaking at 2-3 μm. The mixed mode contains particles with sizes up to 10 μm, with a peak at 4 μm. This information was added in the text.

**p.2 line 7: 'acts to cool the planet'. The direct radiative forcing at TOA by dust is uncertain (as noted by the authors). Kok et al. note that the forcing could actually be positive (leading to warming of the planet). This possibility should be acknowledged.**
**Kok, J.F., D.A. Ridley, Q. Zhou, R.L. Miller, C. Zhao, C.L. Heald, D.S. Ward, S. Albani, and K. Haustein, 2017: Smaller desert dust cooling effect estimated from analysis of dust size and abundance. Nature Geosci., 10, no. 4, 274-278, doi:10.1038/ngeo2912.**

Thank you for the reference. We have acknowledged this new study in the text.

**line 8, 10: replace 'radiative forcing' with 'direct radiative forcing'?**

Replaced.

**p.4 line 15: define 'microphysical distribution' more specifically?**

We modified this sentence to clarify that.

**p.5 line 4: for the benefit of non-specialist readers, define the diameter range of the accumulation mode?**

We added that information in the text: "(nominally 0.1 to 2.5 μm)".

**line 14: 'a technique applied' please provide some description of this technique**

In this part of the text, we want to highlight quantities that were measured and the wavelength range analyzed in different studies. For that reason, we chose not to include details of the techniques used in other studies as that would move away from the main point we would like to make. For clarification, we rewrote the sentence as follows (P5L14):

*"In other studies, aerosol optical properties including the imaginary part of the refractive index and single scattering albedo were determined across a wavelength spectrum from 250 to 800 nm based on measurements of dust particles collected on filters (Muller et al. 2009; Wagner et al. 2012)."*

**Table 1: could you clarify 'Sampling period'? Is each measurement a time-average over this period? Is this period the temporal resolution of each measurement?**

This is clarified now in the last paragraph of the Section 3.2 (P7L4). The sampling period corresponds to the temporal resolution of each instrument. For the LACO Aerosol Sampling Stations, the sampling period of each filter, i.e. the interval on which filters were being replaced, was on the order of hours. Therefore, the dust properties obtained later on based on filter measurements are time-averaged over the sampling period.

**p.8 line 20: 'These peaks are associated with the sudden moistening convective events...' Allen et al (2013) attribute at least part of the event of June 18 to the break-down of a low-level jet (see their Table 1 and their Figure 1c). The same comment applies to p.10 line 9: 'follows a moistening event'.**
**Allen, C. J. T., R. Washington, and S. Engelstaedter (2013), Dust emission and transport mechanisms in the central Sahara: Fennec ground-based observations from Bordj Badji Mokhtar, June 2011, J. Geophys. Res. Atmos., 118, 6212–6232, doi:10.1002/jgrd.50534.**

This is an interesting point. We acknowledged this fact and respective reference in this part of the text (P9L13).

**p.10 line 4: 'On June 22, the detection scale of the nephelometer was reconfigured...' Is this why Figure 4 looks slightly different from the nephelometer times series in Figure 1a of Allen et al. (2013). (It may be unreasonable to expect the authors to know the answer here, but if they do, it would be useful for readers familiar with the FENNEC literature.)**

A possible explanation is that the time series of the scattering coefficient presented in our work passed by several quality controls and fine tuning of the calibration coefficients. The scattering coefficient was also shown on the same time stamps of the reflectometer data.

**p.11 line 8: 'a large range of geometries' What does 'geometries' refer to? The interior of the nephelometer?**

The word 'geometries' was used to refer to the angle between light incidence and detection in the filters. We clarified that in the text (P11L8).

**line 12: 'the reflectance of the Nuclepore filter cannot be increased' Could you put this in plainer language? Do you mean that the filters only get dirtier and less reflective as the air flows through? (For a few events in Figure 5b, the reflectivity at 670 nm initially increases. Is this a measure of instrument error?)**

This slight increase for the red wavelength is indeed related to the uncertainty of the technique. We clarified that in the text (P11L12):

*"This technique is based on the assumptions that the reflectance of the filter decreases as particles are loaded on it, and that the reduction of the light reflected by the filter is due to absorption only by the aerosol particles. The reflectance of the filters for the red wavelength slightly increases when the first particles are collected on their surface. We estimate that this effect is the on order of 1-2%, and it is included in the uncertainties of the reflectance."*

**line 19: what is delta t in this case? 45 seconds, corresponding to the sampling period of the reflectometer? In general, this page is very technical and difficult to read, although instrument scientists will have an easier time.**

The sampling period of the reflectometer was set to 45 seconds, we used $\Delta t = 7.5$ min on the retrieval of the absorption coefficient to have higher sensitivity to the change of filter reflectance. We added this information in the text (P11L26).

**line 25: insert 'temporal' before 'subset'?**

Inserted.

**p.12 line 11: 'These results show variation along this period from 0.96 to close to 1, with a mean value around 0.995...' The low values near 0.96 in Figure 5d are short-lived. Are the values just noise or is there a physical cause?**

The variation in SSA observed is bigger than the instrument noise. In this specific case, the low values of SSA seem to be time correlated to a high dust event. However, we cannot precise the physical cause for that. This might be related to changes in composition or size of the dust particles, but for a complete understanding of physical causes it would be required simultaneous measurements at a high temporal resolution of composition and particles size.

**line 12: 'systematically higher than the values for Saharan dust found in the literature at this wavelength' Please give examples of such studies.**

We moved all the discussion comparing our SSA results with other studies to Sec. 7. References are given there to AERONET, Ryder et al., 2013a, and Otto et al., 2009 (from P26L26 to P27L17).

**Figure 5: Label each panel with a, b, c or d to correspond to the caption.**

The labels were added (Figure 4).

**Figure 6: what do the different colored lines correspond to? Samples from different days?**

Exactly. We show in Figure 5 a set of spectral reflectance of filters collected on different days during the Fennec campaign. We clarified this information in the caption of the figure and in the text (P14L21).

**p.14 line 11 'This strong spectral dependence is what causes the dust to appear brown to our eye.' This is an interesting comment. Could you elaborate by telling me what primary colors brown corresponds to? If I am interpreting the figure correctly, dust absorbs blue and some of green, allowing red to be scattered back to our eyes. (I will always regret not taking an optics class as an undergraduate :)**

Brown is a mix in some specific proportion of red and green/yellow. As you mentioned, most of the red photons are scattered back by dust, while just some of the green/yellow photons are scattered back by these particles (a large fraction of blue light is absorbed by dust). The resulting mix of photons in this proportion is interpreted by our brain as the brown color!

**line 14: 'three groups of samples were identified' Were the Mauritania groups divided based on qualitative inspection or was an objective criterion used?**

We used a qualitative inspection of the shape of the reflectance spectrum of each filter. More specifically, we consider the slope of the reflectance with respect to the wavelength in the spectral range of the visible and the NIR. We added a sentence to better explain the criterion used (P14L26).

*"The reflectance of the filters collected in Mauritania presented significant spectral variations and three groups of samples were identified based on the qualitative inspection of the shape of the reflectance curve in the visible and NIR, as shown in Fig 6."*

**p.15 line 32: 'Other studies also show a decrease in coarse mode fraction as sampling moves towards aged dust and away from fresh dust near sources.' Please be specific and state which of the measurement sites is considered to be closer to the upwind sources.**

Algeria is located more centered at the Saharan desert, closer to the major dust sources. Therefore, the site in Algeria is expected to have higher fraction of fresh dust compared to the site in Mauritania, which is closer to the western coast and receives a higher fraction of transported dust. This information was clarified in the text (P17L8):

*"Other studies also show a decrease in coarse mode fraction as sampling moves towards aged dust and away from fresh dust near the major sources (Weinzierl et al., 2009, 2011; Ansmann et al., 2011; Ryder et al., 2013a, b). This is in agreement that Algeria is more centered in the*

*Saharan desert, closer to the major sources of dust and therefore with higher fraction of fresh dust."*

**p.16 line 3: 'A comparison between this geometrical distribution...' This sentence is unclear. How was the impactor efficiency taken into account in this study? Could the authors explain briefly why this matters? (This is probably obvious to an instrument scientist.)**

In this part of the text we want to highlight that we used a cutoff size of 10um and we have a SEM derived size distribution. Therefore, all the properties were derived for particles on this specific size range. This is important when comparing size-dependent properties with other studies that used different cutoff sampling sizes, or when comparing size distributions derived by optical methods or aerodynamic sizes. We modified this part of the text to clarify that and we added a reference to a paper (Reid et al., 2003), including a detailed comparison of size distributions derived by different techniques (P17L12).

**p.18 line 5: 'Once the instruments arrived back at UMBC, dust deposited on the instrument surfaces was gently collected using a brush and sieved using a 45 µm mesh grid.' Does mechanical brushing break up larger aggregate particles, modifying the aerosol size distribution that was later characterized using the SEM?**

Dust optical properties and size distributions were obtained by the analysis of particles collected on filters. For the grain density, on the other hand, we used a bulk sample of dust. The pycnometer based technique used here for the measurement of density is sensitive to the total mass and volume of the particles as described in [Rocha-Lima et al. 2014]. This measurement does not depend on the particles shape or if they are aggregated or not.

**p.19 line 3: 'Figure 11 a) shows that the imaginary part of the complex refractive index of Saharan dust from Algeria has significant spectral differences between fine and mixed mode.' The authors should explain how the refractive index was calculated for the mixed mode, if the T-matrix code doesn't converge (p.18, line 20). (The answer appears to be in the caption of Figure 11, but this should be described in the text.) For samples from Mauritania, the use of Mie theory seems to introduce an uncertainty comparable to the measurement uncertainty, given the sensitivity of the fine sample index to this assumption. This should be discussed.**

We discussed how the refractive index was calculated for the fine and mixed modes in Section 5.4. For clarity, we added a sentence reinforcing that in this Section 5.5 (P19L12):

*"In the calculation of the absorption efficiency Qabs(m;x), the shape of the fine particles was considered to be first spherical and then spheroidal using Mie theory and T-matrix theory respectively. For the mixed mode, only Mie theory was used since the T-matrix algorithm did not converge for larger particles size."*

As you noted, the retrieval of the imaginary part of the refractive index is more sensitive to the particles shape in Mauritania comparatively to the Algeria case. A possible explanation is the

fact that the absorption efficiency (Qabs) has a sharper variation for smaller size parameters. As you can see in Figure 8b, we have observed a higher fraction of fine particles in the size distribution in Mauritania which could be impacting the retrieval of the mass absorption efficiency and the derivation of the imaginary refractive index. This is now discussed in the text (P20L3):

*"Also for Mauritania, the retrieval of the imaginary part of the refractive index using Mie theory introduced an uncertainty comparable to the uncertainties from the measurements. This is likely related to the presence of more fine particles in this supersite. A possible explanation is the fact that the absorption efficiency (Qabs) has a sharper variation for smaller size parameters, which introduce more variability in the retrieval of the mass absorption efficiency in Eq. 3."*

**line 5: 'For longer wavelength the values diverge considerably...' Why is this? For pure materials, the index of refraction should be independent of particle size. Is this divergence evidence that the fine and mixed samples are comprised of different minerals?**

As seen in Figure 11 and 12, we found differences in the elemental composition between samples of different locations and between fine and mixed samples, but we are unable to correlate composition and optical properties. It should be noted that in addition to composition the crystalline structure might also play a role in the effective refractive index derived for the samples, but we did not further investigate that in this work. Therefore, we cannot precise what is the main reason for the observed differences.

**Figure 12: I find it difficult to derive much quantitative information from this figure. Could the authors replot these four cases as four bar graphs, as in Figure 13?**

As suggested, we replot the EDXRF composition as four bar graphs in Figure 11.

[Figure]

Figure 11. Mean mass concentration in percentage of the total mass of the aerosol particles obtained for each element by Energy Dispersive X-ray Fluorescence analysis (EDXRF) for fine and mixed mode (fine+coarse) of the Saharan dust from Algeria and Mauritania.

**Section 5.6: ('EDXRF analysis of Saharan dust') Are there any differences in elemental composition with respect to particle size? This is an important question for modelers trying to reproduce the hematite composition in soil maps. (e.g. Scanza et al ACP 2015, Perlwitz et al ACP 2015). For example, Perez et al (2016) noted that Fe (from hematite) is mainly independent of particle size at Izana (just downwind of these sites), contrary to some soil mineral atlases that restrict it to larger particle sizes. A figure similar to Figure 13 showing the difference of elemental composition between the fine and mixed modes would be helpful to address this question.**

**Perez Garcia-Pando, C., R. L. Miller, J. P. Perlwitz, S. RodriÌA guez, and J. M. Pros- pero (2016), Predicting the mineral composition of dust aerosols: Insights from ele-.Res.Lett.,43, doi:10.1002/ 2016GL069873. Perlwitz, J. P., Perez Garcia-Pando, C., and Miller, R. L.: Predict- ing the mineral com- position of dust aerosols – Part 1: Representing key processes, Atmos. Chem. Phys., 15, 11593–11627, doi:10.5194/acp-15-11593-2015, 2015. Scanza, R. A., Mahowald, N., Ghan, S., Zender, C. S., Kok, J. F., Liu, X., Zhang, Y., and Albani, S.: Modeling dust as component minerals in the Community Atmosphere Model: development of framework and impact on radiative forcing, Atmos. Chem. Phys., 15, 537– 561, doi:10.5194/acp-15-537-2015, 2015.**

We added two new panels (c) and (d) in Figure 12 with the ratios of fine and mixed elemental concentration for Algeria and Mauritania:

[Figure]

The most notable difference is the higher concentration of Na and Cl (sea salt) and sulfate, vanadium and chrome in the fine fraction in Mauritania. Note that the technique used in our work (EDXRF) is not sensible to the oxidation state of iron or other elements. Although we measured the iron content, we cannot quantify the amount of hematite in these samples without assumptions. We added the following discussion to the text regarding the iron content of the fine and mixed modes (P21L14):

*"The iron content is higher in the mixed mode, although the ratio of fine and mixed concentrations is still compatible with one from the estimated uncertainties. Despite of that, if we consider only the concentration of Fe in Fig. 11, the mean concentration in Algeria (6-7%) is slightly higher than in Mauritania (4.5-5.5%)."*

**p.21 line 15: '...or do the AOT measurements follow the full magnitude of large events.' This could be because cold pools often arrive at night (Allen et al 2013), when the sun photometer is not measuring.**

We have added this information in the text (P23L1). It should be noted however that missing AERONET retrievals due to night time arrivals are not a factor in Figure 13b.

*"In some cases, this could be because cold pools often arrive at night (Marsham et al., 2013; Allen et al., 2013), when the sun photometer is not measuring."*

**p.22 line 5: 'For example, it can be seen in Ryder et al. (2013a) that during fresh dust events...' This is interesting. Here, the authors seem to be arguing that discrepancies between AERONET and the nephelometer are due to departures of the height of the dust layer from the assumed value of 5 km. Can the authors estimate the frequency of this underestimate by AERONET compared to the frequency of missing retrievals because the dust concentration is so high that the aerosol layer is mistaken for a cloud?**

We observed that, at least during the time period that we have measurements (Figure 13), PBL scaled AERONET results departed from our measurements at all times the scattering coefficient was higher than ~400Mm$^{-1}$. In general, the different AERONET data screenings removed most of their measurements from times the AOT was greater than ~3.0 as seen in the figure below.

[Figure]

We see at least three events of high AOD that were present in level 1.0 but not in level 2.0, as highlighted by dashed boxes in the plot. These three events were also seen in our measurements of scattering coefficients as events of high dust concentrations, as showed in Figure 3(a). For a better determination of the frequency of missing retrievals, we would need more statistics and further analysis of the AERONET screening algorithm for the atmospheric conditions during the campaign.

**p.24 line 12: "...follows expected variation associated with sources areas where the sites are located.' Do the authors mean specifically that the Mauritania site is more influenced by marine aerosols?**

We are referring in this part of the text to the association of (Ca +Mg)/Fe and Ca/Al ratios to the location of the dust sources described by Scheuvens et al., 2013 and Formenti et al., 2011, respectively. The ratios obtained for Mauritania and Algeria are in agreement with those previous studies.  Higher ratios were found for samples collected in the Mauritania site located in Western Sahara. We have clarified this part of the text (P25L1):

*"The ratio of some key chemical components, such as Ca/Al and (Ca+Mg)/Al was found larger for the Mauritania site comparatively to the Algeria site which is agreement with previous studies that have linked the location of sources areas and their composition (Formenti et al., 2011; Scheuvens et al., 2013)."*

**p.25 line 21: 'Ryder et al. (2013a) present results of dust optical properties measured and derived during Fennec from aircraft over northern Mauritania and North West of Mali. Differences between these airborne measurements and our ground-based results appear striking at first glance.' SSA depends upon the size distribution (as the authors note below). How much does this contribute to the different compared to possible differences in composition?**

Both the size distribution and the composition of the dust particles contribute to such differences. Ryder (2013a) estimated that adding the full course mode obtained from the aircraft measurements shifts the SSA mean value from 0.97 to 0.92 at 550nm. Moosmuller (2012) estimated that the variation of the concentration of the iron content in different samples results in a SSA variation from 0.99 to 0.86 at 405nm and a variation of 0.999 to 0.992 at 870nm. As our measurements were taken at 640nm, we might have higher sensitivity to particles size than to their composition. However, composition measurements of the airborne measurements would be needed to better disentangle these two effects.

**p.25 line 6: 'However, this bow-shape signature is seen in other previous work by Balkanski et al. (2007) and references therein.' I believe that a bow shape is also implicitly present in some models (e.g. GISS: c.f. Tegen and Lacis 1996; Miller et al JGR 2006) that interpolate the index of refraction between measurements in the visible (Patterson et al 1977 or Sinyuk et al 2003) and IR (Volz 1973). The imaginary part of the index is small within the visible (Sinyuk et al 2003), but rises as a result of interpolation to meet higher values in the IR around 3 um (Volz 1977).**
**Miller, R. L., et al. (2006), Mineral dust aerosols in the NASA Goddard Institute for Space Sciences ModelE atmospheric general circulation model, J. Geophys. Res., 111, D06208, doi:10.1029/2005JD005796.**
**Patterson, E. M., D. A. Gillette, and B. H. Stockton (1977), Complex index of refraction between 300 and 700 nm for Saharan aerosols, J. Geophys. Res., 82, 3153–3160.**
**Sinyuk, A., O. Torres, and O. Dubovik, Combined use of satellite and surface observations to infer the imaginary part of refractive index of Saharan dust, Geophys. Res. Lett., 30(2), 1081, doi:10.1029/ 2002GL016189, 2003.**
**Tegen, I., and A. A. Lacis (1996), Modeling of particle influence on the radiative properties of mineral dust aerosol, J. Geophys. Res., 101, 19,237 – 19,244. Volz, F. E. (1973), Infrared**

**optical constants of ammonium sulfate, Sahara dust, volcanic pumice and fly ash, Appl. Opt., 12, 564–568.**

Thank you for providing these references. We incorporated this information in the text (P26L10).

*"In some models, (e.g. GISS: (Tegen and Lacis, 1996; Miller et al., 2006)), the bow shape is also implicitly present to interpolate measurements of the imaginary part of the refractive index that are small within the visible (Patterson et al., 1977; Sinyuk et al., 2003) to those higher values in the IR (Volz, 1973)."*

**p.26 line 13: 'may correspond to other size-dependent characteristics such as aspect ratio' Or mineral composition? For example, Moosmuller et al (2012) show that for aerodynamic diameters less than 2.5 um, SSA from African dust particles is linearly related to the elemental fraction of iron (which they attribute to hematite).**

**Moosmuller, H., J. P. Engelbrecht, M. Skiba, G. Frey, R. K. Chakrabarty, and W. P. Arnott (2012), Single scattering albedo of fine mineral dust aerosols controlled by iron concentration, J. Geophys. Res., 117, D11210, doi:10.1029/2011JD016909.**

Mineral composition is indeed another important size-dependent characteristic that should be mentioned here. We added that information along with this reference in the manuscript (P27L23):

*"We also find size dependence in the dust absorption spectral signature that has not been noted previously and may correspond to other size-dependent characteristics such as aspect ratio (Kandler et al., 2009) and composition (Kandler et al., 2007; Moosmüller et al., 2012)."*